# Syntheses and Glycosidase Inhibitory Activities, and in Silico Docking Studies of Pericosine E Analogs Methoxy-Substituted at C6

**DOI:** 10.3390/md18040221

**Published:** 2020-04-20

**Authors:** Yoshihide Usami, Megumi Higuchi, Koji Mizuki, Mizuki Yamamoto, Mao Kanki, Chika Nakasone, Yuya Sugimoto, Makio Shibano, Yoshihiro Uesawa, Junko Nagai, Hiroki Yoneyama, Shinya Harusawa

**Affiliations:** 1Department of Pharmaceutical Organic Chemistry, Osaka University of Pharmaceutical Sciences, Nasahara 4-20-1, Takatsuki, Osaka 569-1094, Japan; e14537@gap.oups.ac.jp (M.H.); e12007@gap.oups.ac.jp (K.M.); e13213@gap.oups.ac.jp (M.Y.); e15714@gap.oups.ac.jp (M.K.); e11606@gap.oups.ac.jp (C.N.); e11605@gap.oups.ac.jp (Y.S.); yoneyama@gly.oups.ac.jp (H.Y.); harusawa@gly.oups.ac.jp (S.H.); 2Department of Natural Products Research, Osaka University of Pharmaceutical Sciences, Nasahara 4-20-1, Takatsuki, Osaka 569-1094, Japan; shibano@gly.oups.ac.jp; 3Department of Medical Molecular Informatics, Meiji Pharmaceutical University, 2-522-1 Noshio, Kiyose, Tokyo 204-8588, Japan; uesawa@my-pharm.ac.jp (Y.U.); nagai-j@my-pharm.ac.jp (J.N.)

**Keywords:** pericosine E, marine natural product, C6-methoxy analogue, α-glucosidase inhibitor, anti-diabetes drug, docking simulation

## Abstract

Inspired by the significant α-glucosidase inhibitory activities of (+)- and (−)-pericosine E, we herein designed and synthesized 16 analogs of these marine natural products bearing a methoxy group instead of a chlorine atom at C6. Four of these compounds exhibited moderate α-glucosidase inhibitory activities, which were weaker than those of the corresponding chlorine-containing species. The four compounds could be prepared by coupling reactions utilizing the (−)-pericosine B moiety. An additional in silico docking simulation suggested that the reason of reduced activity of the C6-methoxylated analogs might be an absence of hydrogen bonding between a methoxy group with the surrounding amino acid residues in the active site in α-glucosidase.

## 1. Introduction

Pericosines A–C and E (**1**–**4**) are marine natural products produced by the fungus *Periconia byssoides* OUPS-N133 derived from sea hare *Aplysia kurodai* (Figure 1) [1]. The unique carbasugar structures of these species and the significant antitumor activity of pericosine A have drawn increased attention to pericosines A–C (**1**–**3**) as synthetic targets and thus inspired numerous synthetic studies [2,3,4,5,6,7,8,9]. Natural pericosine E (**4**) deserves particular attention, as it features an *O*-linked carbadisaccharide structure between pericosine A-like and pericosine B-like moieties with opposite absolute configurations and exists as a mixture of enantiomers [1]. In the course of our continuing studies on the total synthesis of marine natural products, we realized the first total syntheses of both enantiomers of **4** and its six stereoisomers, further elucidating their selective α-glucosidase inhibitory activities. The naturally preferred enantiomer (−)-**4** was less active (half maximal (50%) Inhibitory Concentration (IC_50_) = 1.5 × 10^−3^ M) than the minor enantiomer (+)-**4** (IC_50_ = 3.1 × 10^−5^ M), while synthetic analogue (−)-**5** exhibited the highest α-glucosidase inhibitory activity (IC_50_ = 1.2 × 10^−5^ M, ~50 times lower than that of a positive control, deoxynojirimycin). Thus, pericosine E (**4**) is a promising seed for a new class of anti-diabetes drugs [10,11]. 

Considering the above, we herein aimed to design and synthesize pericosine E analogs bearing a methoxy group at C6 instead of a chlorine atom and elucidate the corresponding structure–activity relationships. The design of these targets was inspired by the possible existence of the hitherto unknown naturally occurring pericosine E-type *O*-linked carbadisaccharides comprising known pericosine B or C units with various combinations of chiralities, as the parent pericosine E is thought to be biologically synthesized from pericosines A and B. Moreover, we aimed to determine whether the replacement of the chlorine atom at C6 with a methoxy group influences enzyme inhibitory activity by electronic effect or size of the substituent. In our previous work, the characteristic structural pattern of pericosine E analogs was denoted as (donor, acceptor)-type for a better understanding of the puzzling stereochemistry with eight chiral centers in each compound. For example, (−)-**4** was denoted as (−pA, +pB)-type, as it can be constructed from (−)-pericosine A and (+)-pericosine B moieties. A similar notation was used in the present paper. Inspection of naturally occurring possibilities resulted in the design and synthesis of novel carbadisaccharides **6**, **7**, **8**, and **9**, with (+pC, −pB)-, (+pB, −pB)-, (−pC, −pB)-, and (−pB, −pB)-type structures, respectively. The corresponding enantiomers were also synthesized. 

Herein, we report the syntheses of sixteen pericosine E analogs bearing a methoxy group at C6 instead of a chlorine atom and evaluate their glycosidase inhibitory activities. In addition, results of the docking simulation on active compounds and α-glucosidase is described. 

## 2. Results and Discussion

### 2.1. Syntheses of Pericosine E Analogs Methoxy-Substituted at C6

All possible isomers derived from pericosine B or pericosine C were considered. As mentioned in our previous paper, the acceptor of the coupling reaction was limited to *trans*-epoxide **11** enantiomers [10,11]. (+)-**6**, (−)-**7**, (−)-**8**, and (−)-**9** were prepared following a previously reported strategy (Scheme 1). Both enantiomers of **10** and **15**, which are synthetic precursors of pericosines C and B respectively, could be used as donor molecules in the Lewis-acid-catalyzed coupling reaction [12,13]. 

The synthesis of (+)-**6**, which can be derived from (+)-pericosine C (**3**) and (−)-pericosine B (**2**), is discussed below as an example. BF_3_-catalyzed coupling between (+)-**10** (the precursor of (+)-**3**) and (−)-**11** (the synthetic intermediate of (−)-**2**) yielded (−)-**12** (65%). This compound (i.e., (−)-**12**) was treated with Dess-Martin periodinane (DMP) to afford a crude ketone that was used in the following reaction without purification by silica gel column chromatography to avoid double bond migration. The crude residue was treated with NaBH_4_ in MeOH to afford (+)-**13** (48% yield over two steps from (−)-**12**), which was converted to (+)-**6** in 26% yield by microwave (MW)-assisted acidic deprotection. Alternatively, the epimer (+)-**14** was prepared by similar deprotection of (−)-**12**. Other targets, namely, (−)-**7**–**9** and their epimers (−)-**18**, (−)-**21**, and (−)-**24**, were also prepared according to Scheme 1 under variable conditions. The corresponding enantiomers were synthesized by similar methods to afford sixteen C6-methoxylated pericosine E analogs. Both enantiomers of **10**, **15**, and **11** were prepared as described elsewhere [10,11,12]. The modified selective epoxidation of a cyclohexadiene precursor [10,11] with 1,1,1-trifluorodimethyloxirane to afford *anti*-epoxide **11** is described in the synthesis of the new compound (−)-**15** (see Appendix A).

### 2.2. Evaluation of Glycosidase Inhibitory Activities 

The synthesized compounds were subjected to glycosidase inhibitory activity assays (Table 1). Herein, α-galactosidase (from green coffee bean) and β-galactosidase (from bovine lever) were used in addition to the three previously employed glycosidases (α-glucosidase, β-glucosidase, and α-mannosidase) [11]. To facilitate comparison across studies, we used the same batches of the three abovementioned glycosidases as those employed before. Some compounds were not tested against all five enzymes because of the limited amount of materials. The first three rows of Table 1 relate to (−)- and (+)-pericosine E **4** and the most potent compound (−)-**5** identified in a previous work.

No compounds showed inhibitory activities against β-glucosidase, α-mannosidase, and α-galactosidase, while α-glucosidase was moderately inhibited by (+)-**6**, (−)-**7**, (−)-**8**, and (−)-**24**. The most potent compound, (−)-**24** (IC_50_ = 1.7 × 10^−3^ M), showed an activity close to that of (−)-pericosine E (**4**) (IC_50_ = 1.5 × 10^−3^ M). However, the replacement of a chlorine atom at C6 with a methoxy group decreased activity in general. For example, the α-glucosidase inhibitory activity of (−)-**7** (IC_50_ = 6.2 × 10^−3^ M) was four-fold lower than that of (−)-**4**, which had the same relative and absolute configuration. Moreover, the anti-α-glucosidase activity of (−)-**8** was approximately 1/600 that of (−)-**5**. Similarly, compound (+)-**8** was not active whereas the corresponding (+)-**4** exhibited potent activity. Here, it should be mentioned that these conclusions regarding enzyme activity drawn from the IC_50_ values in this section are approximate.

The above results imply that original pericosine E analogs containing a chlorine atom at C6 are more potent α-glucosidase inhibitors than their methoxy analogues. Notably, compounds featuring (−)-**11** as an acceptor exhibited a certain anti-α-glucosidase activity, except for (−)-**18**. These findings provided useful information for the design of new molecules with better activities.

Gratifyingly, some synthesized compounds, namely, (+)-**9**, (−)-**24**, (+)-**21**, and (+)-**24**, showed β-galactosidase inhibitory activities, featuring (+)-**11** as an acceptor, except for (−)-**24**. The opposite preference for high α-glucosidase and *β*-galactosidase inhibitory activities is interesting and might be valuable for future studies, as the design of new types of β-galactosidase inhibitors is an important and challenging task [14,15,16,17].

### 2.3. Docking Simulation

A molecular docking study using the Dock function [18] of Molecular Operating Environment (MOE) version 2018.0101 (Chemical Computing Group Inc., Quebec, Canada) was conducted for better understanding of the inhibitory mechanisms (Figure 2 and Figure 3). The interactions between α-glucosidase (Protein Data Bank (PDB) code 3A4A) [19] with compounds (−)-**5** and (−)-**24** as well as glucose were investigated using the MOE-Dock.

As a result, the peripheral amino acids important for glucose binding were observed to be involved in the binding of compounds (−)-**5** and (−)-**24**. It was shown that the chlorine atom of compound (−)-**5** could form hydrogen bonds with the surrounding amino acid residues such as Val216, Glu277, Gln279, and Phe303. No such binding was observed for glucose and the methoxy group of the compound (−)-**24**. Furthermore, the configurations of compound (−)-**5** and compound (−)-**24** were quite different, and the surrounding amino acids that bind α-glucosidase were also different.

Tang and co-workers identified 12 amino acid residues, including Val 216, Glu 277, Gln 279, and Phe 303, that interact with the α-glucosidase inhibitors salvianolic acid A and salvianolic acid C. The chlorine-containing compound (−)-**5** might inhibit α-glucosidase in a similar binding manner to those inhibitors [20].

Comparisons of affinity scores between different substrates are known to be useful in assessing agonist/antagonist activity [21]. The binding affinity scores of compounds (−)-**5**, (−)-**24**, and glucose were −6.8, −4.7, and −8.2 kcal/mol, respectively. Glucose, which is included as a partial residue of natural substrates, exhibited the highest affinity among the three docked compounds. Also, the affinities of compounds (−)-**5** and (−)-**24** reflected the inhibitory activities, 1.2 × 10^−5^ and 1.7 × 10^−3^, respectively. These results indicate that the docking studies were achieved with a certain accuracy. On the other hand, the threshold value of the hydrogen bond strength depicted in Figure 2 was set to −1.0 kcal/mol for glucose and −0.1 kcal/mol for derivatives. Compound (−)-**5** is considered to contribute to stronger binding affinity by interacting with more peripheral amino acids than compound (−)-**24**. However, the accuracy of the docking simulation is limited. It is considered necessary to perform molecular dynamics (MD) simulation to obtain more detailed knowledge.

α-glucosidase 3A4A is one of the crystal structures representative of α-glucosidase and is generally used for the study of this enzyme [22,23,24,25]. On the other hand, α-glucosidase, β-glucosidase, α-mannosidase, α-galactosidase, and β-galactosidase used in this study showed different inhibition profiles against various pericosine derivatives. These results suggest that the steric configurations of amino acids at the ligand-binding site among these enzymes are significantly different. The relationship among the amino acid sequences of the ligand-binding sites and the inhibitory activities will be rigorously evaluated in a subsequent study.

## 3. Conclusions

We designed and synthesized 16 pericosine E analogs bearing a methoxy group at C6 and tested their inhibitory activities against five glycosidases. Among the 16 compounds, four exhibited lower α-glucosidase inhibitory activities than the corresponding chlorine-containing species. Thus, the chlorine atom at C6 in pericosine E analogs was concluded to play an important role in determining α-glucosidase inhibitory activity. The identified active compounds generally featured the (−)-pB moiety as an acceptor. Contrarily, some compounds mainly comprising the (+)-pB moiety showed β-galactosidase inhibitory activity, with (−)-**24** acting as a dual inhibitor. Docking simulation suggested that (−)-**24** synthesized in this study binds α-glucosidase in different manner to more potent (−)-**5**, whose chlorine atom forms hydrogen bonds with the surrounding amino acid residues. 

## 4. Experimental

### 4.1. General

Infrared (IR) spectra were recorded on a 1720X Fourier transformation-infrared (FT-IR) spectrometer (Perkin Elmer, MA, USA) or an IRAffinity-1S FT-IR spectrometer (Shimadzu, Kyoto, Japan). High-resolution mass spectra (HRMS) were recorded on a JMS-700 (2) mass spectrometer (JEOL, Tokyo, Japan). Nuclear magnetic resonance (NMR) spectra were recorded at 27 °C on 400-MR-DD2 (Agilent Technologies, CA, USA) and 600-DD2 (Agilent Technologies, CA, USA) instruments in CDCl_3_ or acetone-*d*_6_ using tetramethylsilane as an internal standard. Specific rotations were measured with a JASCO P-2300 spectrometer (JASCO Co., Tokyo, Japan). Liquid column chromatography was conducted on silica gel (Fuji Silysia FL-60D). Analytical thin layer chromatography (TLC) was performed on Silicagel 70 F_254_ plates (Wako Pure Chemical Industries, Tokyo, Japan), and compounds were detected by dipping the plates into an ethanolic solution of phosphomolybdic acid followed by heating. Microwave (MW)-aided reactions were carried out in a Biotage Initiator^®^ reactor (PartnerTech Atvidaberg AB for Biotage Sweden AB, Uppsala, Sweden). NaBH_4_ and trifluoroacetic acid (TFA) were purchased from Wako Pure Chemical Industries (Wako Pure Chemical Industries, Tokyo, Japan). BF_3_·Et_2_O was purchased from Sigma-Aldrich Co., LLC (St. Lewis, MO, USA). DMP was purchased from TCI (Tokyo Chemical Industry Co., Ltd., Tokyo, Japan). (−)-Shikimic acid and (−)-quinic acid were purchased from Carbosynth, Ltd. (UK) and Merck (Merck & Co., Inc., Darmstadt, Germany), respectively. α-Glucosidase (yeast, lot 26010), β-glucosidase (sweet almond, lot 81241), α-mannosidase (Jack Bean, lot 055K7047), α-galactosidase (green coffee bean, lot SMBP1296V), β-galactosidase (bovine liver, lot SMBL0488V), and deoxymannojirimycin were purchased from Sigma-Aldrich Co., LLC (St. Lewis, MO, USA). Deoxygalactonojirimycin was purchased from Funakoshi Co., Ltd., Tokyo, Japan). 1-Deoxynojirimycin (DNJ) was isolated from the leaves of *Morus alba* L. (Procedure for isolation of DNJ is described in Section 4.5.)

### 4.2. Condensation of 10 or 15 with 11

General procedure (synthesis of (−)-**12**, Scheme 1): To a solution of (+)-**10** (68.4 mg, 0.23 mmol) and anti-epoxide (−)-**11** (50.9 mg, 0.15 mmol) in CH_2_Cl_2_ (0.15 mL) was added BF_3_·Et_2_O (3 μL, 0.025 mmol) at 0 °C. After 1 h of stirring, the reaction mixture was treated with Et_3_N (20 µL, 0.14 mmol) and concentrated under vacuum to afford a crude residue that was purified by silica gel column chromatography (eluent = hexane:EtOAc, 1:1 v/v) to afford (−)-**12** (68.8 mg, 65 %) as a colorless oil.

(−)-**12**: oil; ${\left[\alpha \right]}_{D}^{20}$ −7.2 (*c* 1.175, CHCl_3_); IR (film) *ν*_max_ 3438 (OH), 1726 (C=O), 1658 (C=C) cm^−1^; ^1^H-NMR (acetone-*d*_6_, 400 MHz, ppm) δ 1.30−1.70 (20H, m), 3.41 (3H, s, 6’-OMe), 3.73 (3H, s, COOMe), 3.75 (3H, s, COOMe), 4.01–4.04 (1H, m, H-5’), 4.21 (1H, overlapped, H-5), 4.23 (1H, overlapped, H-6), 4.25 (1H, overlapped, H-4’), 4.37 (1H, br d, *J* = 5.4 Hz, H-6’), 4.58 (1H, d, *J* = 3.3 Hz, 5’-OH), 4.70 (1H, ddd, *J* = 6.1, 3.9, 0.9 Hz, H-3’), 4.74 (1H, dd, *J* = 5.6, 2.3 Hz, H-4), 4.78 (1H, ddd, *J* = 5.7, 2.7, 1.2 Hz, H-3), 6.50 (1H, dd, *J* = 2.5, 1.0 Hz, H-2), 6.55 (1H, dd, *J* = 3.7, 1.2 Hz, H-2’); ^13^C-NMR (acetone-*d*_6_, 100 MHz, ppm) δ 24.47 (CH_2_), 24.51 (CH_2_), 24.69 (CH_2_), 24.72 (CH_2_), 25.7 (CH_2_), 25.8 (CH_2_), 35.8 (CH_2_), 36.3 (CH_2_), 37.6 (CH_2_), 38.7 (CH_2_), 52.0 (CH_3_, C-8), 52.1 (CH_3_, C-8’), 60.3 (CH_3_, 6’-OMe), 71.6 (CH, C-3’), 72.5 (CH, C-3), 72.8 (CH, C-5’), 74.0 (CH, C-4), 75.9 (CH, C-6), 76.2 (CH, C-4’), 77.0 (CH, C-6’), 80.5 (C, C-5), 111.1 (Cq), 111.3 (Cq), 133.38 (Cq, C-1), 133.42 (CH, C-2), 134.8 (Cq, C-1’), 136.1 (CH, C-2’), 167.39 (Cq, C-7), 167.43 (Cq, C-7’); high-resolution electron impact mass spectrum (HREIMS) *m/z* calcd for C_29_H_40_O_11_ (M)^+^ 564.2571, found 564.2570.

(+)-**12** (51.2 mg, 90 %) was prepared from (−)-**10** (30 mg, 0.10 mmol) and (+)-**11** (40.0 mg, 0.15 mmol): oil; ${\left[\alpha \right]}_{D}^{20}$ +7.7 (*c* 0.43, CHCl_3_); IR (film) *ν*_max_ 3431 (OH), 1727 (C=O), 1658 (C=C) cm^−1^; ^1^H-NMR (acetone-*d*_6_, 600 MHz, ppm) δ 1.50−1.70 (20H, m), 3.41 (3H, s, 6’-OMe), 3.73 (3H, s, COOMe), 3.75 (3H, s, COOMe), 4.07 (1H, ddd, *J* = 6.8, 5.6, 3.5 Hz, H-5’), 4.21 (1H, dd, *J* = 6.8, 2.9 Hz, H-5), 4.23 (1H, dt, *J* = 6.8, 1.2 Hz, H-6), 4.25 (1H, t, *J* = 6.5 Hz, H-4’), 4.37 (1H, ddd, *J* = 5.6, 1.2, 1.1 Hz, H-6’), 4.60 (1H, d, *J* = 3.5 Hz, OH), 4.70 (1H, ddd, *J* = 6.1, 3.8, 1.1 Hz, H-3’), 4.74 (1H, ddd, *J* = 5.9, 2.9, 0.8 Hz, H-4), 4.78 (1H, ddd, *J* = 5.9, 3.0, 1.5 Hz, H-3), 6.50 (1H, ddd, *J* = 3.0, 1.2, 0.9 Hz, H-2), 6.55 (1H, dd, *J* = 3.8, 1.2 Hz, H-2’); ^13^C-NMR (acetone-*d*_6_, 150 MHz, ppm) δ 24.47 (CH_2_), 24.51 (CH_2_), 24.68 (CH_2_), 24.70 (CH_2_), 25.7 (CH_2_), 25.8 (CH_2_), 35.7 (CH_2_), 36.3 (CH_2_), 37.6 (CH_2_), 38.7 (CH_2_), 52.0 (CH_3_, C-8 or C-8’), 52.1 (CH_3_, C-8 or C-8’), 60.3 (CH_3_, 6’-OMe), 71.6 (CH, C-3’), 72.5 (CH, C-3), 72.7 (CH, C-5’), 74.0 (CH, C-4), 75.9 (CH, C-6), 76.2 (CH, C-4’), 77.0 (CH, C-6’), 80.5 (C, C-5), 111.1 (Cq), 111.3 (Cq), 133.4 (Cq, C-1), 133.5 (CH, C-2), 134.7 (Cq, C-1’), 136.1 (CH, C-2’), 167.40 (Cq, C-7), 167.44 (Cq, C-7’); HRMS *m/z* calcd for C_29_H_40_O_11_ (M)^+^ 564.2571, found 564.2574.

(−)-**16** (150.3 mg, 78 %) was prepared from (+)-**15** (101.4 mg, 0.34 mmol) and (−)-**11** (94.6 mg, 0.36 mmol): oil; ${\left[\alpha \right]}_{D}^{20}$ −75.6 (*c* 0.60, CHCl_3_); IR (film) *ν*_max_ 3477 (OH), 1722 (C=O), 1657 (C=C) cm^−1^; ^1^H-NMR (C_6_D_6_, 300 MHz, ppm) δ 1.00−1.90 (20 H, m), 3.33 (3H, s, H-8), 3.35 (3H, s, H-8’), 3.67 (3H, s, OMe), 4.02 (1H, d, *J* = 4.4 Hz, H-6’), 4.04 (1H, dd, *J* = 21.1, 8.2 Hz, H-5’), 4.16 (1H, dd, *J* = 4.4, 1.2 Hz, H-5), 4.21 (1H, overlapped, H-4’), 4.27 (1H, dd, *J* = 6.5, 3.6 Hz, H-3), 4.35 (1H, ddd, *J* = 6.4, 3.9, 0.5 Hz, H-3’), 4.50 (1H, s, 5’-OH), 4.64 (1H, dd, *J* = 4.4, 1.2 Hz, H-6), 4.66 (1H, dd, *J* = 3.6, 1.5 Hz, H-4), 6.57 (1H, dd, *J* = 3.8, 1.5 Hz, H-2’), 6.79 (1H, d, *J* = 2.9 Hz, H-2); ^13^C-NMR (acetone-*d*_6_, 75 MHz, ppm) δ 24.7 (CH_2_), 24.8 (2 × CH_2_), 25.0 (CH_2_), 25.7 (CH_2_), 26.0 (CH_2_), 36.4 (CH_2_), 36.6 (CH_2_), 38.4 (CH_2_), 38.9 (CH_2_), 52.1 (2 × CH_3_, C-8, 8’), 62.3 (CH_3_, 6’-OMe), 72.0 (CH, C-3’), 72.1 (CH, C-4), 73.3 (CH, C-3), 73.6 (CH, C-6), 74.9 (CH, C-5’),77.1 (CH, C-4’), 78.1 (CH, C-6’), 79.4 (CH, C-5), 112.0 (Cq), 112.9 (Cq), 131.6 (Cq, C-1), 132.9 (CH, C-2’), 136.9 (Cq, C-1’), 136.9 (CH, C-2) 166.8 (Cq, C-7), 166.9 (Cq, C-7’); HREIMS *m/z* calcd for C_29_H_40_O_11_ (M)^+^ 564.2571, found 564.2567.

(+)-**16** (76.2 mg, 90%) was prepared from (−)-**15** (45.0 mg, 0.15 mmol) and (+)-**11** (40.0 mg, 0.15 mmol): oil; ${\left[\alpha \right]}_{D}^{20}$ +75.4 (*c* 0.82, CHCl_3_); IR (film) *ν*_max_ 3488 (OH), 1723 (C=O), 1653 (C=C) cm^−1^; ^1^H-NMR (C_6_D_6_, 600 MHz, ppm) δ 1.50−1.90 (20H, m), 3.36 (3H, s, COOMe), 3.39 (3H, s, COOMe), 3.66 (3H, s, 6’-OMe), 4.02–4.08 (2H, m, H-5’ and H-6’), 4.15 (1H, dd, *J* = 4.7, 3.3 Hz, H-5), 4.21 (1H, ddd, *J* =6.8, 6.7, 2.1 Hz, H-4’), 4.31 (1H, ddd, *J* = 5.8, 3.2, 0.6 Hz, H-3), 4.39 (1H, ddd, *J* = 6.4, 3.8, 0.8 Hz, H-3’), 4.47 (1H, s, 5’-OH), 4.62 (1H, d, *J* = 4.7 Hz, H-6), 4.66 (1H, dd, *J* = 5.9, 2.7 Hz, H-4), 6.59 (1H, dd, *J* = 3.6, 1.4 Hz, H-2’), 6.79 (1H, dd, *J* = 3.2, 0.5 Hz, H-2); ^13^C-NMR (C_6_D_6_, 150 MHz, ppm) δ 24.1 (2 × CH_2_), 24.3 (CH_2_), 24.4 (CH_2_), 25.2 (CH_2_), 25.4 (CH_2_), 35.8 (CH_2_), 36.1 (CH_2_), 37.8 (CH_2_), 38.3 (CH_2_), 51.56 (CH_3_, C-8), 51.60 (CH_3_, C-8’), 61.7 (CH_3_, 6’-OMe), 71.4 (CH, C-3’), 71.5 (CH, C-4), 72.8 (CH, C-3), 73.1 (CH, C-6), 74.2 (CH, C-5’), 76.5 (CH, C-4’), 77.5 (CH, C-6’), 78.9 (CH, C-5), 111.4 (Cq), 112.4 (Cq), 131.1 (Cq, C-1), 132.5 (CH, C-2’), 136.2 (Cq, C-1’), 136.4 (CH, C-2) 166.2 (Cq, C-7), 166.4.(Cq, C-7’); HREIMS *m/z* calcd for C_29_H_40_O_11_ (M)^+^ 564.2571, found 564.2570.

**(**−**)-19** (139 mg, 46%) was prepared from (−)-**10** (187 mg, 0.70 mmol) and (−)-**11** (113 mg, 0.54 mmol): white powder; mp 165–170 °C: ${\left[\alpha \right]}_{D}^{20}$ −48.4 (*c* 0.1, CHCl_3_); IR (KBr) *ν*_max_ 3422 (OH), 1726 (C=O), 1438 (C=C) cm^−1^; ^1^H-NMR (CDCl_3_, 600 MHz, ppm) δ 1.30−1.80 (20H, m), 3.71 (3H, s, 6’-OMe), 3.795 (3H, s, COOMe), 3.796 (3H, s, COOMe), 3.91 (1H, ddd, *J* = 9.4, 7.3, 2.6 Hz, H-5’), 4.05 (1H, dd, *J* = 8.5, 2.3 Hz, H-5), 4.13 (1H, dd, *J* = 9.1, 6.5 Hz, H-4’), 4.25 (1H, ddd, *J* = 7.7, 1.8, 1.4 Hz, H-5’), 4.41 (1H, ddd, *J* = 8.5, 1.8, 1.7 Hz, H-6), 4.61 (1H, overlapped, H-3’), 4.62 (1H, overlapped, H-3), 4.84 (1H, ddd, *J* = 5.6, 2.1, 1.2 Hz, H-4), 4.97 (1H, d, *J* = 2.6 Hz, OH), 6.42 (1H, ddd, *J* = 2.9, 1.8, 1.3 Hz, H-2), 6.60 (1H, dd, *J* = 4.4, 2.0 Hz, H-2’); ^13^C-NMR (CDCl_3_, 150 MHz, ppm) δ 23.6 (CH_2_), 23.8 (CH_2_), 23.92 (CH_2_), 23.95 (CH_2_), 24.9 (CH_2_), 25.0 (CH_2_), 35.1 (CH_2_), 35.8 (CH_2_), 37.2 (CH_2_), 38.1 (CH_2_), 52.0 (CH_3_, C-8 or C-8’), 52.1 (CH_3_, C-8 or C8’), 61.9 (CH_3_, 6’-OMe), 70.7 (CH, C-3), 72.4 (CH, C-3’), 73.4 (CH, C-4), 75.4 (CH, C-5’), 75.7 (CH, C-4’), 76.6 (CH, C-6), 80.0 (CH, C-6’), 82.0 (CH, C-5), 110.8 (Cq), 111.7 (Cq), 130.5 (CH, C-2’), 132.7 (Cq, C-1), 135.4 (CH, C-2), 135.9 (Cq, C-1’), 166.3 (Cq, C-7’), 166.7 (Cq, C-7); HRMS *m/z* calcd for C_29_H_40_O_11_ (M)^+^ 564.2571, found 564.2570. 

**(+)-19** (80 mg, 44%) was prepared from (+)-**10** (86.0 mg, 0.29 mmol) and (+)-**11** (86.2 mg, 0.29 mmol): white powder; mp 165–169 °C: ${\left[\alpha \right]}_{D}^{20}$ +69.0 (*c* 0.75, CHCl_3_); IR (KBr) *ν*_max_ 3408 (OH), 1725 (C=O), 1440 (C=C) cm^−1^; ^1^H-NMR (CDCl_3_, 400 MHz, ppm) δ 1.30−1.80 (20H, m), 3.72 (3H, s, OMe), 3.79 (6H, s, 2 × COOMe), 3.91 (1H, dd, *J* = 8.9, 6.2 Hz), 4.05 (1H, br d, *J* = 8.1 Hz), 4.13 (1H, dd, *J* = 8.4, 5.9 Hz), 4.25 (1H, br d, *J* = 6.5 Hz), 4.41 (1H, br d, *J* = 7.8 Hz), 4.59–4.65 (2H, m), 4.82–4.86 (1H, m), 5.00–5.03 (1H, m), 6.41 (1H, m), 6.60 (1H, m); ^13^C-NMR (CDCl_3_, 100 MHz, ppm) δ 23.6, 23.7, 23.89, 23.92, 24.9, 24.9, 35.1, 35.8, 37.2, 38.1, 52.0, 52.1, 61.9, 70.6, 72.3, 73.4, 75.4, 75.6, 76.6, 80.0, 82.0, 110.8, 111.6, 130.5, 132.6, 135.4, 135.8, 166.2, 166.6; HREIMS *m/z* calcd for C_29_H_40_O_11_ (M)^+^ 564.2571, found 564.2568.

**(**−**)-22** (42.8 mg, 59%) was prepared from (−)-**15** (38.6 mg, 0.13 mmol) and (−)-**11** (45.1 mg, 0.17 mmol): oil; [α]_D_^22.5^ −44.9 (*c* 0.73, CHCl_3_); IR (film) *ν*_max_ 3445 (OH), 1717 (C=O), 1653 (C=C) cm^−1^; ^1^H-NMR (CDCl_3_, 600 MHz, ppm) δ 1.21−1.74 (20H, m), 3.31 (3H, s, OMe), 3.76 (3H, s, COOMe), 3.82 (3H, s, COOMe), 3.87 (1H, br t, *J* = 8.5 Hz, H-5’), 4.07 (1H, dd, *J* = 4.1, 3.0 Hz, H-5), 4.20 (1H, d, *J* = 7.1 Hz, H-6’), 4.21 (1H, dd, *J* = 9.1, 6.8 Hz, H-4’), 4.51–4.53 (1H, m, H-4), 4.65 (1H, ddd, *J* = 6.7, 3.8, 1.1 Hz, H-3’), 4.73 (1H, dd, *J* = 5.9, 3.8 Hz, H-3), 4.82 (1H, s, -OH), 4.83 (1H, d, *J* = 4.1 Hz, H-6), 6.71 (1H, dd, *J* = 3.8, 2.0 Hz, H-2’), 6.97 (1H, dd, *J* = 3.8, 0.9 Hz, H-2); ^13^C-NMR (CDCl_3_, 150 MHz, ppm) δ 23.6 (CH_2_), 23.7 (CH_2_), 23.9 (CH_2_), 25.0 (CH_2_), 35.2 (CH_2_), 35.7 (CH_2_), 37.1 (CH_2_), 37.8 (CH_2_), 52.0 (CH_3_, COOMe), 52.4 (CH_3_, COOMe), 56.1 (CH_3_, 6-OMe), 68.5 (CH, C-6), 70.8 (CH, C-3’), 72.7 (CH, C-3), 73.1 (CH, C-5’), 73.5 (CH, C-4), 76.0 (CH, C-4’), 78.4 (CH, C-5), 78.5 (CH, C-6’), 111.6 (Cq), 112.0 (Cq), 129.1 (Cq, C-1), 132.4 (CH, C-2’), 135.7 (Cq, C-1’), 140.5 (CH, C-2), 166.3 (Cq, C-7’), 167.0 (Cq, C-7); HREIMS *m/z* calcd for C_29_H_40_O_11_ (M)^+^ 564.2571, found 564.2573.

**(+)-22** (41.5 mg, 63%) was prepared from (+)-**15** (34.8 mg, 0.12 mmol) and (+)-**11** (34.3 mg, 0.13 mmol): oil; [α]_D_^21.5^ +51.7 (*c* 0.73, CHCl_3_); IR (film) *ν*_max_ 3474 (OH), 1699 (C=O), 1652 (C=C) cm^−1^; ^1^H-NMR (CDCl_3_, 400 MHz, ppm) δ 1.21−1.71 (20H, m), 3.30 (3H, s, OMe), 3.76 (3H, s, COOMe), 3.82 (3H, s, COOMe), 3.87 (1H, br t, *J* = 8.8 Hz, H-5’), 4.07 (1H, dd, *J* = 4.1, 3.1 Hz, H-5), 4.20 (1H, d, *J* = 6.9 Hz, H-6’), 4.21 (1H, dd, *J* = 9.2, 6.9 Hz, H-4’), 4.52 (1H, dd, *J* =5.7, 2.5 Hz, H-4), 4.66 (1H, br dd, *J* = 6.3, 3.7 Hz, H-3’), 4.74 (1H, dd, *J* = 5.9, 3.7 Hz, H-3), 4.83 (1H, d, *J* = 3.9 Hz, H-6), 4.85 (1H, br s, -OH), 6.72 (1H, dd, *J* = 3.8, 2.0 Hz, H-2’), 6.98 (1H, br d, *J* = 3.5 Hz, H-2); ^13^C-NMR (CDCl_3_, 100 MHz, ppm) δ 23.55, 23.62, 23.89, 23.91, 25.0, 35.2, 35.7, 37.0, 37.8, 52.0, 52.4, 56.1, 68.4, 70.8, 72.7, 73.0, 73.4, 75.9, 78.3, 78.5, 110.6, 112.0, 129.0, 132.4, 135.7, 140.5, 166.2, 166.9; HREIMS *m/z* calcd for C_29_H_40_O_11_ (M)^+^ 564.2571, found 564.2571.

### 4.3. Inversion of Configuration at C5’

General procedure (synthesis of (+)-**13**, Scheme 1): A microwave vial containing a CH_2_Cl_2_ solution (2 mL) of (−)-**12** (47.6 mg, 0.084 mmol) was treated with DMP (77.0 mg, 0.23 mmol) at 0 °C, sealed, and irradiated in the MW reactor at 80 °C for 2.5 h. After cooling, the reaction mixture was treated with aqueous saturated (sat.) Na_2_S_2_O_3_ (20 mL) and saturated aqueous NaHCO_3_ (10 mL) and extracted with *tert*-butyl methyl ether (3 × 20 mL). The combined organic layers were washed with brine (30 mL) and water (30 mL), dried over MgSO_4_, filtered, and concentrated in a vacuum to afford a crude residue containing the desired ketone. To a solution of NaBH_4_ (1.0 mg, 0.026 mmol) in MeOH (0.6 mL), a solution of the crude residue in MeOH (2.0 mL) at 0 °C was added dropwise. After 30 min of stirring, the reaction mixture was treated with saturated aqueous NH_4_Cl (30 mL) and extracted with CH_2_Cl_2_ (3 × 30 mL). The combined organic layers were washed with brine (30 mL), dried over MgSO_4_, filtered, and concentrated in a vacuum to afford the second crude residue, which was purified by silica gel column chromatography (eluent = hexane:EtOAc, 2:1 v/v) to afford **(+)-13** (22.9 mg, 48% from (−)-**12**) as a colorless oil.

**(+)-13**: oil; ${\left[\alpha \right]}_{D}^{20}$ +26.8 (*c* 1.03, CHCl_3_); IR (film) *ν*_max_ 3414 (OH), 1721 (C=O), 1658 (C=C) cm^−1^; ^1^H-NMR (CDCl_3_, 300 MHz, ppm) δ 1.20−1.90 (20H, m), 3.39 (3H, s, 6’-OMe), 3.70–3.80 (1H, overlapped, H-3), 3.75 (3H, s, COOMe), 3.85 (3H, s, COOMe), 4.18 (1H, dd, *J* = 5.9, 3.2 Hz, H-5’), 4.31(1H, d, *J* = 5.9 Hz, H-5), 4.44–4.48 (1H, m, H-6), 4.55 (1H, d, *J* = 4.9 Hz, H-4’), 4.63 (1H, dd, *J* = 5.5, 3.5 Hz, H-6’), 4.68–4.74 (2H, m, H-3’, 4), 4.81 (1H, d, *J* = 11.7 Hz, 5’-OH), 6.72 (1H, br s, H-2), 6.75 (1H, dd, *J* = 3.2, 0.6 Hz, H-2’); ^13^C-NMR (CDCl_3_, 75 MHz, ppm) δ 23.6 (CH_2_), 23.7 (CH_2_), 23.8 (CH_2_), 23.9 (CH_2_), 24.9 (CH_2_), 25.0 (CH_2_), 34.0 (CH_2_), 35.7 (CH_2_), 36.0 (CH_2_), 37.6 (CH_2_), 51.9 (CH_3_, COOMe), 52.3 (CH_3_, COOMe), 59.3 (CH_3_, 6’-OMe), 68.0 (CH, C-5’), 71.5 (CH, C-3), 72.4 (CH, C-3’), 73.2 (2 × CH, C4, C-6’), 74.6 (CH, C-6 or C-4’), 74.7 (CH, C-6 or C-4’), 79.7 (CH, C-5), 110.8 (Cq), 111.7 (Cq), 129.7 (Cq, C-1’), 131.3 (Cq, C-1), 137.5 (CH, C-2 or C-2’), 137.7 (CH, C-2 or C-2’), 166.6 (Cq, C-7 or C-7’), 166.7 (Cq, C-7 or C-7’); HREIMS *m/z* calcd for C_29_H_40_O_11_ (M)^+^ 564.2571, found 564.2570.

**(**−**)-13** (46.3 mg) was prepared in 54% yield from (+)-**12** (85.6 mg, 0.15 mmol): oil; ${\left[\alpha \right]}_{D}^{20}$ −28.3 (*c* 0.745, CHCl_3_); IR (film) *ν*_max_ 3363 (OH), 1720 (C=O), 1655 (C=C) cm^−1^; ^1^H-NMR (CDCl_3_, 600 MHz, ppm) δ 1.36 −1.80 (20 H, m), 3.39 (3H, s, OMe), 3.73–3.75 (1H, m, H-5’), 3.74 (3H, s, COOMe), 3.84 (3H, s, COOMe), 4.18 (1H, dd, *J* = 5.9, 3.5 Hz, H-5’), 4.31(1H, d, *J* = 6.1 Hz, H-5), 4.46 (1H, ddd, *J* = 6.4, 5.6, 1.1 Hz, H-6), 4.55 (1H, d, *J* = 5.0 Hz, H-4’), 4.62 (1H, ddd, *J* = 5.6, 3.2, 0.9 Hz, H-3’), 4.70 (1H, ddd, *J* = 6.4, 3.2, 0.9 Hz, H-3), 4.72 (1H, dd, *J* = 6.4, 3.5 Hz, H-4), 6.73 (1H, d, *J* = 2.3 Hz, H-2), 6.75 (1H, dd, *J* = 3.2, 1.2 Hz, H-2’); ^13^C-NMR (CDCl_3_, 150 MHz, ppm) δ 23.67 (CH_2_), 23.73 (CH_2_), 23.85 (CH_2_), 23.92 (CH_2_), 24.97 (CH_2_), 25.06 (CH_2_), 34.0 (CH_2_), 35.8 (CH_2_), 36.0 (CH_2_), 37.6 (CH_2_), 51.8 (CH_3_, COOMe), 52.3 (CH_3_, COOMe), 59.3 (OMe), 68.1 (CH, C-5’), 71.5 (CH, C-3), 72.4 (CH, C-3’), 73.2 (CH, C-4 or C-6’), 73.3 (CH, C-4 or C-6’), 74.7 (CH, C-6 or C-4’), 74.7 (CH, C-6 or C-4’), 79.6 (CH, C-5), 110.8 (Cq), 111.8 (Cq), 129.8 (Cq, C-1’), 131.3 (Cq, C-1), 137.65 (CH, C-2 or C-2’), 137.70 (CH, C-2 or C-2’), 166.6 (Cq, C-7 or C-7’), 166.7 (Cq, C-7 or C-7’); HRMS *m/z* calcd for C_29_H_40_O_11_ (M)^+^ 564.2571, found 564.2565.

(+)-**17** (34.8 mg, 33%) was prepared from (−)-**16** (106.6 mg, 0.19 mmol): oil; ${\left[\alpha \right]}_{D}^{20}$ +11.7 (*c* 0.34, CHCl_3_); IR (film) *ν*_max_ 3309 (OH), 1718 (C=O), 1655 (C=C) cm^−1^; ^1^H-NMR (acetone-*d*_6_, 300 MHz, ppm) δ 1.20−1.80 (20 H, m), 3.45 (3H, s, 6’-OMe), 3.75 (3H, s, H-8), 3.79 (3H, s, H-8’), 3.91–3.60 (1H, m, H-5’), 4.16 (1H, dd, *J* = 4.1, 2.9 Hz, H-5), 4.20 (1H, br d, *J* = 3.8 Hz, H-6), 4.52 (1H, br d, *J* = 5.3 Hz, H-4’), 4.58 (1H, d, *J* = 3.9 Hz, H-6’), 4.75 (1H, dd, *J* = 5.3, 2.9 Hz, H-3’), 4.82 (1H, ddd, *J* = 6.1, 3.5, 0.6 Hz, H-3), 4.92–4.98 (1H, m, H-4), 6.68 (1H, d, *J* = 3.5 Hz, H-2), 6.77 (1H, d, *J* = 2.7 Hz, H-2’); ^13^C-NMR (acetone-*d*_6_, 75 MHz, ppm) δ 23.5 (CH_2_), 23.6 (CH_2_), 23.7 (CH_2_), 23.9 (CH_2_), 24.8 (2C, CH_2_), 35.2 (CH_2_), 35.4 (CH_2_), 37.2 (CH_2_), 37.5 (CH_2_), 51.3 (CH_3_, C-8), 51.5 (CH_3_, C-8’), 60.5 (CH_3_, 6’-OMe), 71.2 (CH, C-5’), 71.6 (CH, C-6’), 72.3 (CH, C-4), 72.5 (CH, C-3’), 72.8 (CH, C-3), 74.9 (2 × CH, C-6, 4’), 79.2 (CH, C-5), 110.8 (Cq, C-1’), 111.0 (Cq, C-1), 129.4 (Cq), 130.5 (Cq), 136.5 (CH, C-2), 138.2 (CH, C-2’), 165.9 (Cq, C-7), 166.6 (Cq, C-7’); HREIMS *m/z* calcd for C_29_H_40_O_11_ (M)^+^ 564.2571, found 564.2573.

(−)-**17** (24.6 mg, 37%) was prepared from (+)-**16:** oil; ${\left[\alpha \right]}_{D}^{20}$ −14.5 (*c* 0.36, CHCl_3_); IR (film) *ν*_max_ 3518 (OH), 1717 (C=O), 1652 (C=C) cm^−1^; ^1^H-NMR (acetone-*d*_6_, 600 MHz, ppm) δ 1.25−1.70 (20H, m), 3.45 (3H, s, 6’-OMe), 3.75 (3H, s, COOMe), 3.79 (3H, s, COOMe), 3.94 (2H, ddd, *J* = 10.9, 5.0, 2.4 Hz, H-5’), 4.15 (1H, dd, *J* = 4.1, 2.9 Hz, H-5), 4.20 (1H, ddd, *J* = 4.1, 1.2, 0.9 Hz, H-6), 4.52 (1H, dd, *J* = 5.3, 2.4 Hz, H-4’), 4.59 (1H, d, *J* = 5.0 Hz, H-6’), 4.74 (1H, ddd, *J* = 5.3, 2.9, 0.9 Hz, H-3’), 4.81 (1H, ddd, *J* = 6.2, 3.5, 0.6 Hz, H-3), 4.95 (1H, ddd, *J* = 6.1, 2.9, 1.5 Hz, H-4), 6.68 (1H, dd, *J* = 3.5, 0.9 Hz, H-2), 6.77 (1H, dd, *J* = 2.9, 1.2 Hz, H-2’); ^13^C-NMR (acetone-*d*_6_, 150 MHz, ppm) δ 24.5 (CH_2_), 24.58 (CH_2_), 24.65 (CH_2_), 24.8 (CH_2_), 25.75 (CH_2_), 25.78 (CH_2_), 36.1 (CH_2_), 36.3 (CH_2_), 38.2 (CH_2_), 38.4 (CH_2_), 52.2 (CH_3_, C-8), 52.4 (CH_3_, C-8’), 61.4 (CH_3_, 6’-OMe), 72.1 (CH, C-5’), 72.2 (CH, C-6’), 72.5 (CH, C-4), 73.2 (CH, C-3’), 73.4 (CH, C3), 73.7 (CH, C-6), 75.8 (CH, C-4’), 80.2 (CH, C-5), 111.7 (Cq), 111.9 (Cq), 130.4 (Cq, C-1’), 131.6 (Cq, C-1), 137.3 (C, C-2), 139.1 (C, C-2’), 166.8 (Cq, C-7), 167.5 (Cq, C-7’); HREIMS *m/z* calcd for C_29_H_40_O_11_ (M)^+^ 564.2571, found 564.2571.

(−)-**20** (41.8 mg, 48%) was prepared from (−)-**19** (87.2 mg, 0.15 mmol): white powder; ${\left[\alpha \right]}_{D}^{20}$ −23.3 (*c* 0.44, CHCl_3_); IR (KBr) *ν*_max_ 3430 (OH), 1725 (C=O), 1653 (C=C) cm^−1^; ^1^H-NMR (acetone-*d*_6_, 600 MHz, ppm) δ 1.32−1.66 (20H, m), 3.73 (3H, s, 6’-OMe), 3.76 (3H, s, H-8’), 3.78 (3H, s, H-8), 3.88 (1H, ddd, *J* = 12.6, 5.0, 2.9 Hz, H-5’), 4.08 (1H, dd, *J* = 8.5, 2.3 Hz, H-5), 4.22 (1H, dt, *J* = 8.5, 1.7 Hz, H-6), 4.49–4.51 (1H, m, H-4’), 4.63 (1H, d, *J* = 5.0 Hz, H-6’), 4.66 (1H, d, *J* = 12.6 Hz, 5’-OH), 4.72 (1H, ddd, *J* = 5.6, 3.2, 1.2 Hz, H-3’), 4.74 (1H, ddd, *J* = 5.0.2.9,1.7 Hz, H-3), 4.98 (1H, br ddd, *J* = 5.0,2.0,1.5 Hz, H-4), 6.31 (1H, dt, *J* = 3.0, 1.4 Hz, H-2), 6.70 (1H, dd, *J* = 2.9,1.8 Hz, H-2’); ^13^C-NMR (acetone-*d*_6_, 150 MHz, ppm) δ 23.60 (CH_2_), 23.61 (CH_2_), 23.8 (CH_2_), 24.78 (CH_2_), 24.81 (CH_2_), 35.4 (CH_2_), 35.5 (CH_2_), 37.2 (CH_2_), 37.5 (CH_2_), 51.2 (CH_3_, C-8 or C-8’), 51.5 (CH_3_, C-8 or C-8’), 61.8 (CH_3_, 6’-OMe), 68.5 (CH, C-5’), 71.2 (CH, C-6’), 72.30 (CH, C-3), 72.34 (CH, C-3’), 72.9 (CH, C-6), 75.2 (CH, C-4), 77.0 (CH, C-4’), 81.4 (CH, C-5), 109.7 (Cq), 110.8 (Cq), 129.4 (Cq, C-1’), 132.9 (CH, C-1), 135.3 (Cq, C-2), 137.8 (CH, C-2’), 166.1 (Cq, C-7’), 166.7 (Cq, C-7); HRMS *m/z* calcd for C_29_H_40_O_11_ (M)^+^ 564.2570, found 564.2570.

(+)-**20** (44.6 mg, 51%) was prepared from (+)-**19** (79.3 mg, 0.14 mmol): oil; ${\left[\alpha \right]}_{D}^{20}$ +15.6 (*c* 0.27, CHCl_3_); IR (film) ν_max_ 3419 (OH), 1725 (C=O), 1652 (C=C) cm^−1^; ^1^H-NMR (acetone-*d*_6_, 400 MHz, ppm) δ 1.32−1.66 (20H, m), 3.74 (3H, s, OMe), 3.76 (3H, s, C^7^OOMe), 3.78 (3H, s, C^7’^OOMe), 3.88 (1H, ddd, *J* = 12.3, 4.9, 2.9 Hz), 4.10 (1H, dd, *J* = 8.6, 2.4 Hz), 4.22 (1H, dt, *J* = 8.4, 1.8 Hz), 4.48–4.52 (1H, m), 4.63 (1H, d, *J* = 4.7 Hz), 4.67 (1H, d, *J* = 12.5 Hz), 4.71–4.76 (2H, m), 4.98–5.00 (1H, m), 6.30 (1H, br s), 6.70 (1H, dd, *J* = 3.1, 1.0 Hz); ^13^C-NMR (acetone-*d*_6_, 100 MHz, ppm) δ 23.58, 23.59, 23.8, 24.75, 24.78, 35.4, 35.5, 37.2, 37.4, 51.2, 51.4, 61.8, 68.5, 71.1, 72.29, 72.33, 72.8, 75.2, 77.0, 81.4, 109.7, 110.8, 129.3, 132.9, 135.3, 137.8, 166.0, 166.6; HREIMS *m/z* calcd for C_29_H_40_O_11_ (M)^+^ 564.2570, found 564.2573.

**(+)-23** (7.5 mg, 53%) was prepared from (−)-**22** (14.1 mg, 0.025 mmol): oil: ${\left[\alpha \right]}_{D}^{22}$ +24.6 (*c* 0.26, CHCl_3_); IR (film) *ν*_max_ 3445 (OH), 1722 (C=O), 1437 (C=C) cm^−1^; ^1^H-NMR (CDCl_3_, 600 MHz, ppm) δ 1.29−1.71 (20H, m), 3.44 (3H, s, OMe), 3.87 (1H, br t, *J* = 3.5 Hz, H-5’), 3.78 (3H, s, COOMe), 3.82 (3H, s, COOMe), 3.99 (1H, dd, *J* = 4.7, 3.0 Hz, H-5), 4.35 (1H, dd, *J* = 5.9, 3.0 Hz, H-4), 4.57 (1H, dd, *J* = 5.2, 3.2 Hz, H-4’), 4.62 (1H, dd, *J* = 5.3, 3.5 Hz, H-3), 4.64 (1H, d, *J* = 5.3 Hz, H-6’), 4.72 (1H, dd, *J* = 5.9, 3.5 Hz, H-3’), 4.76 (1H, d, *J* = 4.4 Hz, H-6), 6.86 (1H, dd, *J* = 3.5, 0.9 Hz, H-2), 6.93 (1H, dd, *J* = 3.8, 0.9 Hz, H-2’); ^13^C-NMR (CDCl_3_, 150 MHz, ppm) δ 23.66 (CH_2_), 23.73 (CH_2_), 23.8 (CH_2_), 23.9(CH_2_), 25.0 (CH_2_), 35.4 (CH_2_), 35.5 (CH_2_), 37.2 (CH_2_), 52.25 (CH_3_, COOMe), 52.31 (CH_3_, COOMe), 57.9 (CH_3_, 6-OMe), 68.4 (CH, C-5’), 69.3 (CH, C-6), 70.9 (CH, C-6’), 72.2 (CH, C-3’), 72.6 (CH, C-3), 73.8 (CH, C-4), 74.6 (CH, C-4’), 76.2 (CH, C-5), 110.85 (Cq), 111.88 (Cq), 129.8 (Cq, C-1), 130.4 (Cq, C-1’), 139.0 (CH, C-2’), 139.0 (CH, C-2), 166.9 (Cq, C-7’), 167.1 (Cq, C-7); HREIMS *m/z* calcd for C_29_H_40_O_11_ (M)^+^ 564.2571, found 564.2567.

(−)**-23** (5.5 mg, 25%) was prepared from (+)-**22** (22.1 mg, 0.039 mmol): oil: ${\left[\alpha \right]}_{D}^{20}$ −29.7 (*c* 0.26, CHCl_3_); IR (film) *ν*_max_ 3495 (OH), 1717 (C=O), 1436 (C=C) cm^−1^; ^1^H-NMR (CDCl_3_, 400 MHz, ppm) δ 1.25−1.72 (20H, m), 3.44 (3H, s, OMe), 3.75–3.79 (1H, m, H-5’), 3.78 (3H, s, COOMe), 3.82 (3H, s, COOMe), 3.99 (1H, dd, *J* = 4.3, 2.9 Hz, H-5), 4.35 (1H, dd, *J* = 5.6, 2.7 Hz, H-4), 4.58 (1H, dd, *J* = 5.6, 3.2 Hz, H-4’), 4.62–4.65 (2H, overlapped, H-3, H-6’), 4.72 (1H, dd, *J* = 5.9, 3.6 Hz, H-3’), 4.76 (1H, d, *J* = 3.4 Hz, H-6), 6.86 (1H, br d, *J* = 3.0 Hz, H-2), 6.94 (1H, br d, *J* = 3.5 Hz, H-2’); ^13^C-NMR (CDCl_3_, 100 MHz, ppm) δ 23.66, 23.74, 23.8, 23.9, 25.0, 35.4, 35.5, 37.2, 52.29, 52.35, 57.9, 68.4, 69.3, 70.8, 72.2, 72.7, 73.8, 74.6, 76.1,110.9, 111.9, 111.9, 129.8, 130.4, 139.0, 139.1, 166.9, 167.1; HREIMS *m/z* calcd for C_29_H_40_O_11_ (M)^+^ 564.2571, found 564.2573.

### 4.4. Deprotection to Carbadisaccharides

General procedure (synthesis of (+)-**6** from (+)-**13**, Scheme 1): A microwave vial containing a methanolic solution (0.5 mL) of (+)-**13** (36.0 mg, 0.064 mmol) was charged with TFA (4.5 mL) at 0 °C, sealed, and irradiated in the MW reactor at 80 °C for 5.5 h. After cooling, the reaction mixture was concentrated in a vacuum to afford a crude residue that was purified by silica gel column chromatography (eluent = MeOH:CH_2_Cl_2_, 2:8 v/v) to afford (+)-**6** (6.6 mg, 26%).

(+)-**6**:oil; ${\left[\alpha \right]}_{D}^{20}$ +57.8 (*c* 0.37, EtOH); IR (film) *ν*_max_ 3384 (OH), 1718 (C=O), 1653 (C=C) cm^−1^; ^1^H-NMR (acetone-*d*_6_, 400 MHz, ppm) δ 3.48 (3H, s, 6’-OMe), 3.75 (1H, overlapped, H-5’) 3.75 (3H, s, COOMe), 3.80 (3H, s, COOMe), 3.95(1H, br d, *J* = 3.1 Hz, H-4’), 4.03 (1H, br s, H-4’), 4.13 (1H, br d, *J* = 4.7 Hz, H-3), 4.22 (1H, overlapped, H-3’), 4.24 (1H, br d, *J* = 2.4 Hz, H-5), 4.35 (1H, d, *J* = 3.1 Hz, H-6), 4.52 (1H, d, *J* = 3.9 Hz, H-6’), 6.69 (1H, br s, H-2’), 6.88 (1H, d, *J* = 4.5 Hz, H-2); ^13^C-NMR (acetone-*d*_6_, 100 MHz, ppm) δ 52.1 (CH_3_, C-8), 52.4 (CH_3_, C-8’), 58.6 (CH_3_, 6’-OMe), 65.9 (CH, C-3), 68.0 (CH, C-4), 69.2 (CH, C-3’), 70.5 (CH, C-5’), 72.4 (CH_,_ C-4’), 76.2 (CH, C-6), 76.8 (CH, C-6’), 82.7 (CH, C-5), 130.0 (Cq, C-1), 130.4 (Cq, C-1’), 141.7 (CH, C-2), 142.7 (CH, C-2’), 167.0 (Cq, C-7’), 167.5 (Cq, C-7); HRFABMS *m/z* calcd for C_17_H_25_O_11_Na (M + Na)^+^ 427.1216, found 427.1216.

(+)-**14** (4.2 mg, 22%) was prepared from (−)-**12** (27.0 mg, 0.048 mmol): white powder; ${\left[\alpha \right]}_{D}^{20}$ +27.5 (*c* 0.095, EtOH); IR (film) *ν*_max_ 3361 (OH), 1724 (C=O), 1657 (C=C) cm^−1^; ^1^H-NMR (acetone-*d*_6_, 600 MHz, ppm) δ 3.48 (3H, s, 6’-OMe), 3.73 (3H, s, COOMe), 3.76 (3H, s, COOMe), 3.88 (1H, ddd, *J* = 9.7, 5.0, 2.4 Hz, H-4), 3.94 (1H, br dd, *J* = 10.0, 4.7 Hz, H-4’), 4.04 (1H, ddd, *J* = 10.5, 5.3, 5.0 Hz, H-3), 4.18 (1H, d, *J* = 4.7 Hz, H-5), 4.22 (1H, br d, *J* = 2.9 Hz, H-6’), 4.25 (1H, d, *J* = 7.3 Hz, -OH), 4.35 (1H, d, *J* = 10.7 Hz, -OH), 4.40 (1H, d, *J* = 3.2 Hz, H-6), 4.42 (1H, br ddd, *J* = 7.3, 3.5, 3.2 Hz, H-3’), 4.47 (1H, ddd, *J* = 5.6, 3.8, 3.5 Hz, H-5’), 4.67 (1H, d, *J* = 3.8 Hz, -OH) 6.73 (1H, br d, *J* = 2.9 Hz, H-2’), 6.87 (1H, d, *J* = 4.7 Hz, H-2); ^13^C-NMR (acetone-*d*_6_, 150 MHz, ppm) δ 52.1 (2 × CH_3_, C8, C8’), 58.8 (CH_3_, 6’-OMe), 66.3 (CH, C-3), 66.7 (CH, C-3’), 67.2 (CH, C-4), 69.6 (CH, C-5’), 70.3 (CH, C-4’), 76.0 (CH, C-6), 76.8 (CH, C-6’),81.2 (CH, C-5), 130.4 (Cq, C-1’), 130.5 (Cq, C-1), 141.3 (CH, C-2), 141.5 (CH, C-2’), 167.0 (Cq, C-7’), 167.5 (Cq, C-7); HRFABMS *m/z* calcd for C_17_H_25_O_11_Na (M + Na)^+^ 427.1216, found 427.1216.

**(**−**)-6** (8.4 mg, 58%) was prepared from (−)-**13** (20.2 mg, 0.036 mmol): white powder; ${\left[\alpha \right]}_{D}^{20}$ −61.5 (*c* 0.22, EtOH); IR (film) *ν*_max_ 3392 (OH), 1718 (C=O), 1653 (C=C) cm^−1^; ^1^H-NMR (acetone-*d*_6_, 600 MHz, ppm) δ 3.47 (3H, s, 6’-OMe), 3.73 (1H, overlapped, H-5’) 3.74 (3H, s, COOMe), 3.80 (3H, s, COOMe), 3.94(1H, dd, *J* = 5.0, 1.8 Hz, H-4’), 4.03 (1H, br s, H-4’), 4.12 (1H, br dd, *J* = 5.0, 4.7 Hz, H-3), 4.21 (1H, overlapped, H-3’), 4.22 (1H, dd, *J* = 3.0, 2.3 Hz, H-5), 4.33 (1H, d, *J* = 3.2 Hz, H-6), 4.51 (1H, d, *J* = 4.1 Hz, H-6’), 6.69 (1H, d, *J* = 1.2 Hz, H-2’), 6.87 (1H, d, *J* = 4.7 Hz, H-2); ^13^C-NMR (acetone-*d*_6_, 150 MHz, ppm) δ 52.1 (CH_3_, C-8), 52.4 (CH_3_, C-8’), 58.6 (CH_3_, 6’-OMe), 66.0 (CH, C-3), 68.0 (CH_,_ C-4), 69.2 (CH_,_ C-3’), 70.5 (CH_,_ C-5’), 72.4 (CH_,_ C-4’), 76.2 (CH, C-6), 76.8 (CH, C-6’), 82.7 (CH, C-5), 130.1 (Cq, C-1), 130.5 (Cq, C1’), 141.7 (CH, C-2), 142.6 (CH, C-2’), 167.0 (Cq, C-7’), 167.5 (Cq, C-7); HRFABMS *m/z* calcd for C_17_H_25_O_11_ (M + H)^+^ 405.1397, found 405.1399.

(−)-**14** (11.1 mg, 40%) was prepared from (+)-**12** (38.4 mg, 0.068 mmol): white powder; ${\left[\alpha \right]}_{D}^{20}$ −27.6 (*c* 0.13, EtOH); IR (film) *ν*_max_ 3415 (OH), 1720 (C=O), 1655 (C=C) cm^−1^; ^1^H-NMR (acetone-*d*_6_, 600 MHz, ppm) δ 3.50 (3H, s, 6’-OMe), 3.74 (3H, s, COOMe), 3.78 (3H, s, COOMe), 3.92 (1H, dd, *J* = 5.0, 2.4 Hz, H-4), 3.96 (1H, br dd, *J* = 5.0, 4.7 Hz, H-4’), 4.08 (1H, dd, *J* = 5.0, 4.7 Hz, H-3), 4.15 (1H, dd, *J* = 2.9, 2.7 Hz, H-5), 4.22 (1H, d, *J* = 2.9 Hz, H-6’), 4.41 (1H, d, *J* = 3.2 Hz, H-6), 4.43 (1H, br dd, *J* = 3.3, 2.9 Hz, H-3’), 4.47 (1H, dd, *J* = 5.6, 3.2 Hz, H-5’), 6.75 (1H, d, *J* = 2.6 Hz, H-2’), 6.88 (1H, d, *J* = 4.7 Hz, H-2); ^13^C-NMR (acetone-*d*_6_, 150 MHz, ppm) δ 52.2 (CH_3_, C-8), 52.3 (CH_3_, C-8’), 58.8 (CH_3_, 6’-OMe), 66.1 (CH, C-3), 66.5 (CH, C-3’), 67.1 (CH, C-4), 69.2 (CH, C-5’), 70.0 (CH, C-4’), 76.0 (CH, C-6), 76.6 (CH, C-6’), 80.9 (CH, C-5), 130.3 (Cq, C-1’), 130.4 (Cq, C-1), 141.4 (CH, C-2), 141.7 (CH-C-2’), 167.2 (Cq, C-7’), 167.6 (Cq, C-7); HRFABMS *m/z* calcd for C_17_H_25_O_11_ (M + H)^+^ 405.1397, found 405.1391.

(−)-**7** (6.2 mg, 54%) was prepared from (+)-**17** (16.1 mg, 0.029 mmol): oil; ${\left[\alpha \right]}_{D}^{20}$ −10.5 (*c* 0.10, EtOH); IR (film) *ν*_max_ 3355 (OH), 1717 (C=O), 1652 (C=C) cm^−1^; ^1^H-NMR (acetone-*d*_6_, 400 MHz, ppm) δ 3.47 (3H, s, 6’-OMe), 3.77 (3H, s, COOMe), 3.79 (3H, s, COOMe), 3.82–3.86 (1H, m, H-5’), 3.88–4.04 (2H, m, OH, OH), 3.98–4.04 (1H, m, H-4’), 4.09 (1H, br d, *J* = 2.2 Hz, H-5), 4.18–4.28 (3H, m, OH, H-3’, H-3), 4.32 (1H, d, *J* = 3.7 Hz, H-6), 4.43–4.49 (1H, m, H-4), 4.67 (1H, d, *J* = 3.7 Hz, H-6’), 4.66–4.74 (1H, br, OH), 6.73 (1H, d, *J* = 2.2 Hz, H-2), 6.82 (1H, d, *J* = 2.0 Hz, H-2’); ^13^C-NMR (acetone-*d*_6_, 100 MHz, ppm) δ 51.6 (CH_3_, C-8), 51.7 (CH_3_, C-8’), 60.9 (CH_3_, 6’-OMe), 68.6 (CH, C-3), 68.9 (CH, C-3’), 69.1 (CH, C-5’), 70.2 (CH, C-4), 72.2 (CH, C-4’), 75.0 (CH, C-6’), 75.4 (CH, C-6), 79.6 (CH, C-5), 129.5 (CH, C-1),129.7 (CH, C-1’),141.4 (CH, C-2), 142.4 (CH, C-2’), 165.9 (Cq, C-7), 166.4 (Cq, C-7’); HREIMS *m/z* calcd for C_17_H_25_O_11_ (M + H)^+^ 405.1397, found 405.1395.

(−)-**18** (5.0 mg, 41%) was prepared from (−)-**16** (17.0 mg, 0.030 mmol): oil; *R_f_* = 0.16 (MeOH:CH_2_Cl_2_ = 1:9 v/v); ${\left[\alpha \right]}_{D}^{20}$ −21.6 (*c* 0.275, EtOH); IR (film) *ν*_max_ 3357 (OH), 1718 (C=O), 1657 (C=C) cm^−1^; ^1^H-NMR (acetone-*d*_6_, 400 MHz, ppm) δ 3.31 (1H, d, *J* = 5.3 Hz, -OH), 3.49 (3H, s, 6’-OMe), 3.70–3.78 (1H, m, H-4’), 3.777 (3H, s, COOMe), 3.779 (3H, s, COOMe), 3.82–3.88 (2H, m, -OH, -OH), 3.99 (1H, dd, *J* = 3.7, 2.2 Hz, H-5), 4.07 (1H, d, *J* = 7.0 Hz, -OH), 4.15 (1H, d, *J* = 8.0 Hz, -OH), 4.20–4.28 (2H, overlapped, H-5’, H-3), 4.32–4.40 (3H, overlapped, H-6, H-6’, H-3’), 4.42–4.48 (1H, m, H-4), 4.56 (1H, d, *J* = 3.3 Hz, OH) 6.74 (1H, dd, *J* = 2.4, 1.4 Hz, H-2), 6.77 (1H, d, *J* = 3.3 Hz, H-2’); ^13^C-NMR (acetone-*d*_6_, 100 MHz, ppm) δ 51.56 (CH_3_, C-8), 51.61 (CH_3_, C-8’), 61.1 (CH_3_, 6’-OMe), 65.9 (CH_2_, C-3’), 69.1 (CH_2_, C-3), 70.3 (CH_2_, C-4), 70.4 (CH_2_, C-4’), 71.0 (CH_2_, C-5’), 75.5 (CH_2_, C-6), 77.7 (CH, C-6’), 79.4 (CH, C-5), 129.2 (CH, C-1), 131.8 (CH, C-1’), 139.1 (CH, C-2’), 141.8 (CH, C-2), 165.8 (Cq, C-7), 166.8 (Cq, C-7’); HREIMS *m/z* calcd for C_17_H_25_O_11_ (M + H)^+^ 405.1397, found 405.1399.

(+)-**7** (6.8 mg, 52%) was prepared from (−)-**17** (20.1 mg, 0.036 mmol): oil; ${\left[\alpha \right]}_{D}^{20}$ +11.4 (*c* 0.11, EtOH); IR (film) *ν*_max_ 3359 (OH), 1717 (C=O), 1652 (C=C) cm^−1^; ^1^H-NMR (acetone-*d*_6_, 400 MHz, ppm) δ 3.47 (3H, s, 6’-OMe), 3.77 (3H, s, COOMe), 3.79 (3H, s, COOMe), 3.84 (1H, ddd, *J* = 8.2, 4.1, 2.2 Hz, H-5’), 3.89 (1H, d, *J* = 7.4 Hz, -OH), 3.94 (1H, d, *J* = 9.7 Hz, OH), 4.00 (1H, d, *J* = 9.8 Hz, -OH), 3.98–4.04 (1H, m, H-4’), 4.09 (1H, dd, *J* = 3.8, 2.3 Hz, H-5), 4.19 (1H, d, *J* = 8.8 Hz, -OH), 4.22–4.28 (2H, m, H-3’ and H-3), 4.32 (1H, d, *J* = 4.0 Hz, H-6), 4.43–4.49 (1H, m, H-4), 4.68 (1H, d, *J* = 8.3 Hz, -OH), 4.67 (1H, d, *J* = 4.1 Hz, H-6’), 6.73 (1H, dd, *J* = 2.6, 4.3 Hz, H-2), 6.83 (1H, dd, *J* = 2.5, 1.3 Hz, H-2’); ^13^C-NMR (acetone-*d*_6_, 100 MHz, ppm) δ 51.6 (CH_3_, C-8), 51.7 (CH_3_, C-8’), 60.9 (CH_3_, 6’-OMe), 68.6 (CH, C-3), 68.9 (CH, C-3’), 69.1 (CH, C-5’), 70.3 (CH, C-4), 72.2 (CH, C-4’), 75.0 (CH, C-6’), 75.4 (CH, C-6), 79.6 (CH, C-5), 129.5 (Cq, C-1), 129.7 (Cq, C-1’), 141.3 (CH, C-2), 142.4 (CH, C-2’), 165.9 (Cq, C-7), 166.4 (Cq, C-7’); HREIMS *m/z* calcd for C_17_H_24_O_11_ (M)^+^ 404.1319, found 404.1315.

(+)-**18** (7.1 mg, 62%) was prepared from (+)-**16** (16.1 mg, 0.029 mmol): oil; ${\left[\alpha \right]}_{D}^{20}$ +24.3 (*c* 0.36, EtOH); IR (liquid film) *ν*_max_ 3361 (OH), 1718 (C=O), 1655 (C=C) cm^−1^; ^1^H-NMR (acetone-*d*_6_, 400 MHz, ppm) δ 3.31 (1H, d, *J* = 5.1 Hz, -OH), 3.49 (3H, s, 6’-OMe), 3.70–3.78 (1H, m, H-4’), 3.776 (3H, s, COOMe), 3.778 (3H, s, COOMe), 3.86 (1H, br d, *J* = 9.5 Hz, -OH), 3.99 (1H, dd, *J* = 3.7, 2.0 Hz, H-5), 4.07 (1H, br s, -OH), 4.14 (1H, br d, *J* = 8.2 Hz, -OH), 4.20–4.28 (2H, overlapped, H-5’, H-3), 4.32–4.40 (3H, overlapped, H-6, H-6’, H-3’), 4.42–4.48 (1H, m, H-4), 4.55 (1H, br s, -OH) 6.74 (1H, dd, *J* = 2.2, 1.4 Hz, H-2), 6.77 (1H, br d, *J* = 33 Hz, H-2’); ^13^C-NMR (acetone-*d*_6_, 100 MHz, ppm) δ 51.56 (CH_3,_ C-8), 51.61 (CH_3,_ C-8’), 61.1 (CH_3_, 6’-OMe), 65.9 (CH_2_, C-3’), 69.1 (CH_2_, C-3), 70.4 (2 × CH_,_ C-4, C-4’), 71.0 (CH_,_ C-5’), 75.5 (CH_,_ C-6), 77.7 (CH, C-6’), 79.4 (CH, C-5), 129.2 (Cq, C-1), 131.8 (Cq, C-1’), 139.1 (CH, C-2’), 141.8 (CH, C-2), 165.8 (Cq, C-7), 166.8 (Cq, C-7’); HREIMS *m/z* calcd for C_17_H_24_O_11_ (M)^+^ 404.1319, found 404.1317.

**(**−**)-8** (10.3 mg, 45%) was prepared from (−)-**20** (31.8 mg, 0.056 mmol): oil; ${\left[\alpha \right]}_{D}^{20}$ −96.7 (*c* 0.48, EtOH); IR (film) *ν*_max_ 3419 (OH), 1728 (C=O), 1653 (C=C) cm^−1^; ^1^H-NMR (acetone-*d*_6_, 600 MHz, ppm) δ 3.32 (3H, s, 6-OMe), 3.59–3.78 (1H, m, H-5’), 3.76 (3H, s, COOMe), 3.81 (3H, s, COOMe), 3.96–3.99 (2H, m, H-4, H-4’), 4.13 (1H, br t, *J* = 4.1 Hz, H-3), 4.33 (1H, dd, *J* = 4.1, 1.8 Hz, H-5), 4.54 (1H, d, *J* = 4.1 Hz, H-6), 4.58 (1H, *J* = 4.2 Hz, H-6’), 6.83 (1H, dd, *J* = 2.3, 1.5 Hz, H-2’), 6.85 (1H, d, *J* = 4.1 Hz, H-2); ^13^C-NMR (acetone-*d*_6_, 150 MHz, ppm) δ 52.1 (CH_3_, COOMe), 52.5 (CH_3_, COOMe), 59.0 (CH_3_, -OMe), 66.7 (CH, C-3), 67.8 (CH, C-4), 69.1 (CH, C-3’), 69.7 (CH, C-5’), 72.3 (CH, C-4’), 73.4 (CH, C-6’), 76.5 (CH, C-6), 80.6 (CH, C-5), 130.1 (Cq, C-1’), 130.9 (Cq, C-1), 141.7 (CH, C-2), 143.3 (CH, C-2’), 167.4 (Cq, C-7), 167.5 (Cq, C-7); HRMS *m/z* calcd for C_17_H_25_O_11_ (M + H)^+^ 405.1397, found 405.1390.

**(**−**)-21** (10.3 mg, 45%) was prepared from (−)-**20** (31.8 mg, 0.056 mmol): oil; ${\left[\alpha \right]}_{D}^{20}$ −215.4 (*c* 0.35, EtOH); IR (film) *ν*_max_ 3393 (OH), 1716 (C=O), 1653 (C=C) cm^−1^; ^1^H-NMR (acetone-*d*_6_, 600 MHz, ppm) δ 3.35 (3H, s, 6-OMe), 3.75 (1H, dd, *J* = 6.7, 3.5 Hz, H-4’), 3.77 (3H, s, COOMe), 3.780 (3H, s, COOMe), 4.00 (2H, dd, *J* = 4.7, 2.0 Hz, H-4), 4.08 (1H, dd, *J* = 6.7, 4.1 Hz, H-5’), 4.14 (1H, dd, *J* = 4.7, 4.1 Hz, H-3), 4.28 (1H, dd, *J* = 4.1, 2,1 Hz, H-5), 4.35 (1H, d, *J* = 3.8 Hz, H-6’), 4.37 (1H, t, *J* = 3.8 Hz, H-3’), 4.42 (1H, br d, *J* = 4.1 Hz, H-6’), 4.35 (1H, *J* = 3.8 Hz, H-6’), 4.42 (1H, d, *J* = 3.8 Hz, H-6), 4.35 (1H, *J* = 3.8 Hz, H-6’), 4.64 (1H, d, *J* = 7.1 Hz, -OH), 6.81 (1H, d, *J* = 3.5 Hz, H-2’), 6.83 (1H, d, *J* = 4.1 Hz, H-2); ^13^C-NMR (acetone-*d*_6_, 150 MHz, ppm) δ 52.2 (CH_3_, COOMe), 52.4 (CH_3_, COOMe), 59.2 (CH_3_, -OMe), 66.4 (CH, C-3’), 66.9 (CH, C-3), 67.9 (CH, C-4), 70.7 (CH, C-4’), 71.8 (CH, C-5’), 76.66 (CH, C-6’), 76.70 (CH, C-6), 81.1 (CH, C-5), 130.8 (Cq, C-1), 131.1 (Cq, C-1’), 141.1 (CH, C-2’), 141.5 (CH, C-2), 167.6 (Cq, C-7), 167.9 (Cq, C-7’); HRMS *m/z* calcd for C_17_H_25_O_11_ (M + H)^+^ 405.1397, found 405.1393.

**(+)****-8** (10.5 mg, 33%) was prepared from (+)-**20** (44.6 mg, 0.079 mmol): oil; ${\left[\alpha \right]}_{D}^{20}$ +86.9 (*c* 0.43, EtOH); IR (film) *ν*_max_ 3383 (OH), 1716 (C=O), 1652 (C=C) cm^−1^; ^1^H-NMR (acetone-*d*_6_, 400 MHz, ppm) δ 3.51 (3H, s, -OMe), 3.76 (3H, s, COOMe), 3.80 (3H, s, COOMe), 3.95–4.00 (2H, m, H-4, H-4’), 4.10–4.14 (2H, m), 4.24–4.28 (1H, m), 4.32 (1H, dd, *J* = 4.3, 2.0 Hz, H-5), 4.52 (1H, d, *J* = 4.3 Hz, H-6), 4.59 (1H, d, *J* = 4.3 Hz, H-6’), 6.81 (1H, d, *J* = 1.6 Hz, H-2’), 6.84 (1H, d, *J* = 4.3 Hz, H-2); ^13^C-NMR (acetone-*d*_6_, 100 MHz, ppm) δ 50.7, 51.1, 57.6, 65.3, 66.4, 67.7, 68.3, 70.9, 72.0, 75.1, 79.3, 128.7, 129.4, 140.3, 142.0, 165.9, 166.1; HREIMS *m/z* calcd for C_17_H_25_O_11_ (M + H)^+^ 405.1397, found 405.1393.

**(+)****-21** (9.0 mg, 32%) was prepared from (+)-**19** (40.0 mg, 0.07 mmol): oil; ${\left[\alpha \right]}_{D}^{20}$ +211 (*c* 0.095, EtOH); IR (film) *ν*_max_ 3419 (OH), 1716 (C=O), 1653 (C=C) cm^−1^; ^1^H-NMR (acetone-*d*_6_, 400 MHz, ppm) δ 3.49 (3H, s, 6-OMe), 3.76 (3H, s, COOMe), 3.79 (3H, s, COOMe), 3.83 (1H, br d, *J* = 7.8 Hz) 3.90–3.39 (2H, m), 4.04–4.14 (2H, m), 4.19–4.24 (1H, m), 4.28 (1H, dd, *J* = 3.9, 2.9 Hz), 4.36 (1H, d, *J* = 4.1 Hz), 4.36–4.39 (1H, m), 4.41 (1H, d, *J* = 3.9 Hz, H-6’), 4.53–4.57 (1H, m), 6.80 (1H, d, *J* = 3.3 Hz), 6.82 (1H, d, *J* = 4.3 Hz); ^13^C-NMR (acetone-*d*_6_, 100 MHz, ppm) 52.2, 52.4, 59.2, 66.4, 66.9, 67.8, 70.7, 71.8, 76.6, 76.7, 81.1, 130.7, 131.1, 141.0, 141.1, 167.5, 167.9 HREIMS *m/z* calcd for C_17_H_25_O_11_ (M)^+^ 405.1396, found 405.1390.

**(**−**)-24** (1.9 mg, 26%) was prepared from (−)-**22** (10.2 mg, 0.018 mmol): oil; ${\left[\alpha \right]}_{D}^{20}$ −20.0 (*c* 0.05, EtOH); IR (film) *ν*_max_ 3403 (OH), 1700 (C=O), 1653 (C=C) cm^−1^; ^1^H-NMR (acetone-*d*_6_, 600 MHz, ppm) δ 3.56 (3H, s, OMe), 3.70 (1H, dt, *J* = 7.0, 4.4 Hz, H-4’), 3.72 (1H, d, *J* = 10.2 Hz, -OH), 3.77 (3H, s, COOMe), 3.78 (3H, s, COOMe), 3.92 (1H, d, *J* = 6.5 Hz, -OH), 3.94 (1H, d, *J* = 7.9 Hz, -OH), 4.04 (1H, dd, *J* = 3.8, 2.1 Hz, H-6), 4.10–4.14 (1H, m, H-3), 4.14 (1H, d, *J* = 7.1 Hz, OH), 4.23 (1H, br dt, *J* = 7.9, 4.4 Hz, H-5’), 4.35 (1H, br dt, *J* = 7.0, 4.1 Hz, H-3’), 4.38 (1H, d, *J* = 3.8 Hz, 3’-OH), 4.40 (1H, d, *J* = 4.7 Hz, H-6’), 4.53 (1H, d, *J* = 3.8 Hz, H-6), 6.70 (1H, dd, *J* = 2.6, 1.1 Hz, H-2), 6.72 (1H, dd, *J* = 4.1, 0.9 Hz, H-2’); ^13^C-NMR (acetone-*d*_6_, 150 MHz, ppm) 52.1 (CH_3_, -COOMe), 52.2 (CH_3_, -COOMe), 61.1(CH_3_, -OMe), 66.6 (CH, C-3’), 69.5 (CH, C-3), 70.2 (CH, C-4), 71.1 (CH, C-4’), 71.9 (CH, C-5’), 76.7 (CH, C-6), 78.0 (CH, C-6’), 79.0 (CH, C-5), 130.6 (Cq, C-1), 132.7 (Cq, C-1’), 139.3 (CH, C-2’), 141.8 (CH, C-2), 166.8 (Cq, C-7), 167.6 (Cq, C-7’); HREIMS *m/z* calcd for C_17_H_24_O_11_ M^+^ 404.1318, found 404.1311.

**(**−**)-9** (2.8 mg, 43%) was prepared from (+)-**23** (9.1 mg, 0.016 mmol): oil; ${\left[\alpha \right]}_{D}^{20}$ −49.0 (*c* 0.085, CHCl_3_); IR (film) *ν*_max_ 3420 (OH), 1719 (C=O), 1655 (C=C), 1507 (C=C), 1458 (C=C) cm^−1^; ^1^H-NMR (CDCl_3_, 600 MHz, ppm) δ 3.56 (3H, s, OMe), 3.81–3.85 (2H, m, overlapped), 3.819 (3H, s, COOMe), 3.824 (3H, s, COOMe), 3.93 (1H, br s, -OH), 3.94 (1H, d, *J* = 2.6 Hz, -OH), 4.02–4.04 (1H, m), 4.14–4.17 (1H, m), 4.22–4.23 (1H, m), 4.25 (1H, dd, *J* = 3.5, 2.1 Hz), 4.35 (1H, m, OH), 4.51 (1H, d, *J* = 3.5 Hz), 4.63 (1H, d, *J* = 4.4 Hz), 6.88 (1H, d, *J* = 4.1 Hz), 6.94 (1H, dd, *J* = 3.2, 0.9 Hz); ^13^C-NMR (CDCl_3_, 150 MHz, ppm) 52.3, 52.5, 59.2, 67.3, 67.8, 69.0, 69.2, 70.5, 73.4, 74.9, 78.7, 129.4, 129.9, 140.2, 141.3, 166.4, 166.7; HREIMS *m/z* calcd for C_17_H_22_O_10_ (M − H_2_O)^+^ 386.1213, found 386.1210.

**(****+)-24** (5.3 mg, 38%) was prepared from (+)-**22** (14.1 mg, 0.023 mmol): oil; ${\left[\alpha \right]}_{D}^{20}$ +20.7 (*c* 0.29, EtOH); IR (film) *ν*_max_ 3400 (OH), 1702 (C=O), 1652 (C=C) cm^−1^; ^1^H-NMR (acetone-*d*_6_, 400 MHz, ppm) δ 3.56 (3H, s, OMe), 3.71 (1H, dd, *J* = 7.4, 4.1 Hz, H-4’), 3.77 (3H, s, COOMe), 3.78 (3H, s, COOMe), 4.07 (1H, dd, *J* = 3.7, 1.8 Hz, H-6), 4.14 (1H, br s, -OH), 4.18 (1H, br s, -OH), 4.23 (1H, dd, *J* = 7.4, 4.8 Hz, H-5’), 4.35 (1H, br t, *J* = 4.1 Hz, H-3’), 4.40 (1H, d, *J* = 4.8 Hz, H-6’), 4.53 (1H, d, *J* = 3.7 Hz, H-6), 6.70 (1H, dd, *J* = 2.6, 1.0 Hz, H-2), 6.72 (1H, br d, *J* = 3.9 Hz, H-2’); ^13^C-NMR (acetone-*d*_6_, 100 MHz, ppm) 52.07, 52.13, 61.1, 66.5, 69.5, 70.1, 71.0, 71.8, 76.5, 78.0, 79.0, 130.5, 132.6, 139.3, 141.8, 166.7, 167.6; HREIMS *m/z* calcd for C_17_H_22_O_10_ (M − H_2_O)^+^ 386.1213, found 386.1209.

**(+)-9** (5.5 mg, 73%) was prepared from (−)-**23** (10.7 mg, 0.019 mmol): oil; ${\left[\alpha \right]}_{D}^{20}$ +49.5 (*c* 0.17, CHCl_3_); IR (film) *ν*_max_ 3410 (OH), 1719 (C=O), 1655 (C=C), 1506 (C=C), 1456 (C=C) cm^−1^; ^1^H-NMR (CDCl_3_, 400 MHz, ppm) δ 3.58 (3H, s, OMe), 3.78–3.84 (2H, m, overlapped), 3.82 (3H, s, COOMe), 3.83 (3H, s, COOMe), 3.90–3.96 (1H, m), 4.40–4.18 (1H, m), 4.12–4.18 (1H, m), 4.20–4.26 (1H, m), 4.29 (1H, dd, *J* = 3.3, 1.8 Hz), 4.44 (1H, br s, OH), 4.47 (1H, d, *J* = 3.7 Hz), 4.63 (1H, d, *J* = 4.1 Hz), 6.87 (1H, d, *J* = 3.9 Hz), 6.93 (1H, br d, *J* = 2.9 Hz,); ^13^C-NMR (CDCl_3_, 100 MHz, ppm) 52.3, 52.5, 59.2, 67.1, 67.9, 68.8, 69.2, 70.5, 74.6, 75.1, 78.5, 129.4, 130.1, 139.7, 141.4, 166.6, 166.7; HREIMS *m/z* calcd for C_17_H_25_O_11_ (M + H)^+^ 405.1396, found 405.1395.

### 4.5. Assays of Glycosidase Inhibitory Activity 

#### Assay of α-Glucosidase Inhibitory Activity

The assay reaction mixture comprised 0.1 M acetate buffer (pH 5.0, 45 μL), 20 mM *p*-nitrophenyl-α-d-glucopyranoside solution (25 μL), and α-glucosidase solution (25 μL, stock solution of 1.0 mg/mL in 50 mM Tris-HCl-buffer at pH 7.8 diluted 20-fold with 10 mM phosphate buffer at pH 7.0), with the test samples or DNJ (5 μL solution, concentration range 0.1–20 mg/mL). After 20 min incubation at 37 °C, the reaction was quenched by addition of Na_2_CO_3_ solution (0.5 M, 100 μL). The amount of liberated *p*-nitrophenol was measured colorimetrically at 400 nm (optical density at 400 nm: ODtest). Inhibition efficiencies (%) were calculated as 100 − 100 × (ODtest − ODblank)/(control ODtest − control ODblank), and IC_50_ values (Table 1) were obtained from inhibition curves. 

Assays of β-glucosidase, α-mannosidase, β-mannosidase, and β-galactosidase inhibitory activities were carried out as above using *p*-nitrophenyl-β-d-glucopyranoside, *p*-nitrophenyl-α-d-mannopyranoside, *p*-nitrophenyl-β-d-mannopyranoside, and *p*-nitrophenyl-β-d-galactopyranoside as substrates. The corresponding IC_50_ values are listed in Table 1.

*Isolation of positive control deoxynojirimycin: Dried leaves of *Morus alba* L. (0.5 kg) were cut finely and then extracted with hot water (10 L) for 2 h. The extracted solution was chromatographed on an Amberlite CG-50 (H^+^-form) column (6.5 mm inside diameter (i.d.) × 30 cm length). After washing the column with water and then 50 % MeOH, the adsorbed material was eluted with 50 % MeOH-28 % ammonia solution (9:1). The eluted fraction was concentrated in vacuo to give a basic fraction (5.0 g). This fraction was chromatographed on a Dowex 50W-X4 column (200–400 mesh, 5.0 mm i.d. × 20 cm ) pretreated with formic acid-ammonium formate buffer (0.2 M ammonia formate, adjusted to pH 5.7 with 1 M formic acid), with stepwise elution (H_2_O, H_2_O-28% ammonia solution (99:1, 9:1)). The fraction (H_2_O-28 % ammonia solution (99:1)) was re-chromatographed on semi-preparative HPLC (column:Shodex NH_2_P (4.6 mm i.d. × 250 mm), solvent:CH_3_CN-H_2_O (80:20), flow rate: 1.0 mL/min, column temperature: ambient). 1-Deoxynojirimycin (40 mg) was finally obtained.

### 4.6. Docking Simulation 

The docking analysis was carried out using α-glucosidase protein (PDB code 3A4A) using the Dock induced-fit function in Molecular Operating Environment (MOE) version 2018.0101 (Chemical Computing Group Inc., Quebec, Canada) to better understand the inhibitory mechanisms. [18,19] The calculation of the binding affinity scoring function was performed with the amber 10:eht force field, triangle matcher as placement, and GBVI/WSA dG as the binding affinity scoring function [26]. In the protein preparation with respect to charged residues in the binding site, the Protonate three dimensional (3D) option in MOE was used to determine the ionization states and add hydrogens to the system [27]. The function of the Protonate 3D allows to assign ionization states and position hydrogens in a macromolecular structure given its 3D coordinates from the crystal structure. Hydrogen atoms are required for all atom molecular mechanics, dynamics, or electrostatic calculations. The addition of hydrogen atoms to a macromolecule is a non-trivial task; generally, one must determine the rotamers of -SH -OH -CH_3_ and -NH_3_ groups in cysteine (CYS), serine (SER), tyrosine (TYR), threonine(THR), methionine (MET), and lysine (LYS), the ionization states of acids and bases in arginine (ARG), aspartic acid (ASP), glutamic acid (GLU), LYS, histidine (HIS), the tautomers of imidazoles (HIS) and carboxylic acids (ASP, GLU), the protonation state of metal-ligand atoms CYS, HIS, ASP, GLU, etc., and the ionization state of metals.

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
