# Peer review of "Syntheses and Glycosidase Inhibitory Activities, and in Silico Docking Studies of Pericosine E Analogs Methoxy-Substituted at C6"

_marinedrugs, 2020, doi:10.3390/md18040221_

Round 1
Reviewer 1 Report
Usami et al. describe the synthesis and biological evaluation of pericosine E analogues with a methoxy substituent at C6, in place of the natural chloride. They report 16 new compounds, which either have reduced or no activity against alpha-glucosidase. QSAR analysis suggested that the biological activity correlated with 3D shape descriptors of the molecules.
There is no reason to dispute any of the results presented, and the experimental data is clearly presented. However, there is not much novelty in this paper at the moment. The authors cite their own previous publication in the same journal from 2017, which outlined both a new synthetic procedure AND a significantly more potent analogue. As there are no major synthetic developments nor improvements in potency, I believe that the work described is not sufficiently interesting for publication in Marine Drugs in its present form. Furthermore the QSAR analysis is somewhat simplistic (it is not really surprising that 3D shape affects activity of any bioactive small molecule). I would recommend that the authors consider synthesising more analogues with increased structural variants before resubmitting this paper as well as describing their QSAR analysis in a lot more detail. At the moment the descriptors are simply pasted in a table with minimal explanation.
Author Response
Reply to Reviewer 1
Suggestion 1: I would recommend that the authors consider synthesising more analogues with increased structural variants before resubmitting this paper.
Response 1: We are sorry that we cannot satisfy the reviewer’s requirement at this moment. Each synthesis of designed compound was so delicate from stereochemical combination that it is equal to total synthesis of natural products. It took several years by our hands for synthesizing designed 16 stereoisomers.
Suggestion 2: Furthermore the QSAR analysis is somewhat simplistic (it is not really surprising that 3D shape affects activity of any bioactive small molecule).  describing their QSAR analysis in a lot more detail. At the moment the descriptors are simply pasted in a table with minimal explanation.
Response 2: In-silico study was removed from other reviewers’ suggestion.
Reviewer 2 Report
Well written manuscript focused on synthesis and structure activity relationship of nove pericosine derivatives - compounds with interesting structure resembling sugars that may be used for selective blockade of the alpha-glucosidase. This enzyme is involved in diabetes, which means the topic is worth studying. Authors describe in detail synthesis and testing against various glycosidases, hitting mostly the alpha-glucosidase (selectivity) and also beta-galactosidase, although the affinities are rather weak (mostly millimolar range).
My major concern is the QSAR study which does not improve the manuscript quality, it actually makes it less acceptable for the computationally oriented scientific public. QSAR analyses using huge amount of descriptors with small numbers of data points (case here) inevitably lead to chance correlations - some descriptors will always be correlated with the biological activity. However, these can neither explain properly observed effects, nor be prospectively used for designing new, improved compounds.
The "correlations" found by authors are very weak and I cannot agree with their statement that they could explain the structure-activity relationships using some of the 3D descriptors.
I would recommend the authors to remove completely the QSAR part and replace it with a structure-based approach. My short research in the PDB showed that there might be suitable structures to which authors could try to dock (e.g. using AutoDock) their compounds (e.g. PDB ID: 5NN4 = Crystal structure of human lysosomal acid-alpha-glucosidase, GAA, in complex with N-acetyl-cysteine; maybe use Uniprot to see other glucosidases of different origin). These would it turn provide mechanistic insights and lead to true structure-activity relationships or at least trends, in any case making the manuscript much more solid.
Author Response
Reply to Reviewer 2
Suggestion 1: My major concern is the QSAR study which does not improve the manuscript quality, it actually makes it less acceptable for the computationally oriented scientific public. QSAR analyses using huge amount of descriptors with small numbers of data points (case here) inevitably lead to chance correlations - some descriptors will always be correlated with the biological activity. However, these can neither explain properly observed effects, nor be prospectively used for designing new, improved compounds.The "correlations" found by authors are very weak and I cannot agree with their statement that they could explain the structure-activity relationships using some of the 3D descriptors.
Response 1: Thank you for suggestion. I removed in-silico study from Title and text, experimental and references.
Suggestion 2: I would recommend the authors to remove completely the QSAR part and replace it with a structure-based approach. My short research in the PDB showed that there might be suitable structures to which authors could try to dock (e.g. using AutoDock) their compounds (e.g. PDB ID: 5NN4 = Crystal structure of human lysosomal acid-alpha-glucosidase, GAA, in complex with N-acetyl-cysteine; maybe use Uniprot to see other glucosidases of different origin). These would it turn provide mechanistic insights and lead to true structure-activity relationships or at least trends, in any case making the manuscript much more solid.
Response 2: Thank you for suggestion. We have no enough time to prepare docking study. The reviewer’s suggestion will be done in our future study.
Reviewer 3 Report
The paper describes the design, synthesis, and glycosidase inhibitory activity of sixteen pericosine E analogs. Although the research is very meaningful, several issues of the article make me not to recommend it for publication at this time.
There are no definite criteria indeed, but IC50 at micromolar level concentration is definitely too high for an enzyme inhibitor to be considered good. Moreover, novel analogs are much less potent than the lead compounds against glycosidase. Also, the controls data were quoted from the reference, not obtained from the same bench assay with the novel analogs in in vitro enzyme inhibition test.
For regression analysis, a high r2 of above 0.60 is required, otherwise it could not be reasonably predicted to some degree of accuracy in the science research.
And a lot of errors: such as "α-glycosidase" "IC50 = 2.5" (Page 2, line 2); "(-)-pericosine B (2)" (Page 3, line 6); "ehibited" (Page 4, line 16); structure of (+)-26 (Page 5, Figure 2) etc.
Author Response
Suggestion 1: There are no definite criteria indeed, but IC50 at micromolar level concentration is definitely too high for an enzyme inhibitor to be considered good. Moreover, novel analogs are much less potent than the lead compounds against glycosidase. Also, the controls data were quoted from the reference, not obtained from the same bench assay with the novel analogs in in vitro enzyme inhibition test.
Response 1: Synthesized compounds showed less activity than analogs with chlorine atom as we mentioned in text. Then, important role of choline atom in a series of analogs could be confirmed from results of this work.
Suggestion 2: For regression analysis, a high r2 of above 0.60 is required, otherwise it could not be reasonably predicted to some degree of accuracy in the science research.
Response 2: Description of in-silico study was removed from text.
Suggestion 3: And a lot of errors: such as "α-glycosidase" "IC50 = 2.5" (Page 2, line 2); "(-)-pericosine B (2)" (Page 3, line 6); "ehibited" (Page 4, line 16); structure of (+)-26 (Page 5, Figure 2) etc.
Response 3: We checked and revised. Thank you very much.
Reviewer 4 Report
This article clearly describes the synthesis of pericosin E analogs methoxy-substituted at C-6 and attempts to evaluate their glycosidase inhibitory activities. The newly designed analogs are well characterized by optical rotation, IR, NMR and HRMS.
I have the following comments and questions:
Description of Figure 1: The compound 5 is not mentioned. Does it belong to natural pericosines or to newly designed analogues?
Table 1: The last two lines are somehow shifted in the table. Why the positive control for alpha-galactosidase was not measured?
Paragraph 4.1, last sentence-there should be some citation/description of the method about the isolation of deoxynojirimycin from the leaves.
Paragraph 4.1-A spectrometer where the optical density was measured should be mentioned.
Paragraph 2.3 QSAR analysis- the conclusion (ii) that “the 3D structure of molecules contributes to the alpha-glucosidase inhibitory activity …” is generally known for any enzyme and inhibitor/substrate.
Do the results of alpha-galactosidase inhibitory activities obtained experimentally (Table 1) correspond to the results obtained from the QSAR analysis? Could you comment on that in the text, for me it is not clear.
The IC50 determination is sufficient to approximate the inhibitory properties of the substrate (inhibitor) but it says nothing about the real enzyme-inhibitor binding (this method is suitable when you are working with cells, for example). In order to monitor the inhibitory properties of the substrate against a particular enzyme, the authors must measure the inhibition constant. Please add/replace IC50 with inhibition constants in Table 1 (and than comment on that) otherwise, the conclusions about the inhibitory properties of the substrates are too vague.
Author Response
Reply to Reviewer 4
Question 1: Description of Figure 1: The compound 5 is not mentioned. Does it belong to natural pericosines or to newly designed analogues?
Response: Compound 1 is not a new compound seen in our previous paper. Some description can be seen in page 1 and 4 line 7.
Question 2: Table 1: The last two lines are somehow shifted in the table. Why the positive control for alpha-galactosidase was not measured?
Response 2: It was not measured because all synthesized compounds were inactive against a-galactosidase.
Question 3: Paragraph 4.1, last sentence-there should be some citation/description of the method about the isolation of deoxynojirimycin from the leaves.
Response 3: Description was added in Experimental part.
Question 4: Paragraph 4.1-A spectrometer where the optical density was measured should be mentioned.
Response 4: We added information.
Question 5: Paragraph 2.3 QSAR analysis- the conclusion (ii) that “the 3D structure of molecules contributes to the alpha-glucosidase inhibitory activity …” is generally known for any enzyme and inhibitor/substrate.
Do the results of alpha-galactosidase inhibitory activities obtained experimentally (Table 1) correspond to the results obtained from the QSAR analysis? Could you comment on that in the text, for me it is not clear.
Response 5: In silico study part was removed from other reviewers’ suggestions.
Question 6: The IC50 determination is sufficient to approximate the inhibitory properties of the substrate (inhibitor) but it says nothing about the real enzyme-inhibitor binding (this method is suitable when you are working with cells, for example). In order to monitor the inhibitory properties of the substrate against a particular enzyme, the authors must measure the inhibition constant. Please add/replace IC50 with inhibition constants in Table 1 (and than comment on that) otherwise, the conclusions about the inhibitory properties of the substrates are too vague.
Response 6: We are sorry that we cannot provide binding constants in this time. The reviewer’s comment will be accepted our continuing study. Thank you for suggestion.
Round 2
Reviewer 1 Report
I am glad that the authors removed the QSAR analysis from the discussion. Please also remove it from the title of the paper. After that, the paper should be suitable for publication.
Author Response
I am glad that the authors removed the QSAR analysis from the discussion. Please also remove it from the title of the paper. After that, the paper should be suitable for publication.
Response: Thank you for advice.
Reviewer 2 Report
Authors removed the problematic part focused on the QSAR analysis of the ligands, but unfortunately did not add the part including structure-based modeling. Thus in this form the article now present very few new insights on both in vitro (novel compounds do not show too much biological activity) and structural biology (e.g. possible modes of action, binding modes etc.).
In general, I do not think the current version reaches the quality standards of the Marine Drugs Journal.
Author Response
Authors removed the problematic part focused on the QSAR analysis of the ligands, but unfortunately did not add the part including structure-based modeling. Thus in this form the article now present very few new insights on both in vitro (novel compounds do not show too much biological activity) and structural biology (e.g. possible modes of action, binding modes etc.).
In general, I do not think the current version reaches the quality standards of the Marine Drugs Journal.
Response: docking study was performed and it was added in text.
Reviewer 3 Report
No more comments except the description of the activity, IC50 with mM level concentration as significant
Author Response
No more comments except the description of the activity, IC50 with mM level concentration as significant.
Response: “significant” was change to “moderate”.
Reviewer 4 Report
I thank the authors for their answers, most of my questions were answered satisfactorily. But I have to insist on the inhibition constants (question 6). From our own experience in the laboratory, we known that Ki value is not always approximately 1/3 of the IC50 as sometimes misrepresented in the literature, Ki values may also be of an order of magnitude different from IC50. There is no defined relationship between IC50 an Ki, therefore, enzyme inhibitory activities cannot be inferred from IC50 values. If the article is published without inhibition constants, it would be good if the authors at least mentioned that they were aware of this fact (that is, that the conclusions regarding enzyme inhibitory activities are only approximate).
Author Response
Reviewer's comments:I thank the authors for their answers, most of my questions were answered satisfactorily. But I have to insist on the inhibition constants (question 6). From our own experience in the laboratory, we known that Ki value is not always approximately 1/3 of the IC50 as sometimes misrepresented in the literature, Ki values may also be of an order of magnitude different from IC50. There is no defined relationship between IC50 an Ki, therefore, enzyme inhibitory activities cannot be inferred from IC50 values. If the article is published without inhibition constants, it would be good if the authors at least mentioned that they were aware of this fact (that is, that the conclusions regarding enzyme inhibitory activities are only approximate).
Response: Thank you for useful advice. But we consumed all samples for bioassay to determine IC50, so we cannot present KIs in this paper as mentioned in previous response. Since designed molecules have structural similarity with glucose, so we think these molecules shows the activity by competitive inhibition and mechanistic elucidation of inhibition type was not aimed in this paper. In addition, we show some papers published in Marine Drugs with bioactive molecules expressed in only IC50 values and without KIs. Paper by Chen and co-workers concerns on alpha-glucosidase inhibition with IC50 value.
- Du et al., Lysophosphatidylcholines and Chlorophyll-Derived Molecules from the Diatom Cylindrotheca closterium with Anti-Inflammatory Activity Drugs 2020, 18(3), 166; https://doi.org/10.3390/md18030166
- Cheng et al., Terpenoids from the Deep-Sea-Derived Fungus Penicillium thomii YPGA3 and Their Bioactivities Drugs 2020, 18(3), 164; https://doi.org/10.3390/md18030164
- Feng et al., Sesquiterpenes and Cyclodepsipeptides from Marine-Derived Fungus Trichoderma longibrachiatum and Their Antagonistic Activities against Soil-borne Pathogens Drugs 2020, 18(3), 165; https://doi.org/10.3390/md18030165
Round 3
Reviewer 2 Report
Authors followed my suggestion and improved the story by including some structure-activity relationships based on molecular docking. However, I still think the paper has far more potential for improvement, here some of the points that should be addressed:
- is the binding site of PDB ID 3A4A (isomaltase) homologous to the active sites of the beta-glucosidase, alpha-mannosidase, and alpha-
galactosidase, and alpha-glucosidase? If there are differences in amino acids, this has to be presented and discussed. - The study does not need "some" docking, I would like to see a fully detailed analysis of the predicted binding modes. The figures as they appear now do not provide enough insight into how it actually all works. Authors should reduce the zone of visible residues in the active site to the minimum needed for correct understanding of the structure-activity relationships, 2D figures are messy and do not help too much either.
- I would like to see the correlation of the docked pose scores with the experimental inhibition data measured by the authors or similar compounds measured by other groups.
- the docking protocol should be described in more detail, especially protein preparation with respect to charged residues in the binding site that have a crucial impact on hydrogen bonding and thus on the docked poses and their scoring. According to my pKa prediction program, the GLU 277 residue seems to be protonated in 3A4A which is quite special and may have interesting consequences for docking and activity studies... What authors saw in their case?
Author Response
c
Comment#1: Authors followed my suggestion and improved the story by including some structure-activity relationships based on molecular docking. However, I still think the paper has far more potential for improvement, here some of the points that should be addressed: is the binding site of PDB ID 3A4A (isomaltase) homologous to the active sites of the beta-glucosidase, alpha-mannosidase, and alpha-galactosidase, and alpha-glucosidase? If there are differences in amino acids, this has to be presented and discussed.
Response #1: According to reviewer's comment, we corrected on line 12 in page 7 as “alpha-glucosidase 3A4A is one of the crystal structures representative of alph-glucosidase and is generally used for the study of this enzyme[22-25]. On the other hand, alpha-glucosidase, beta-glucosidase, alpha-mannosidase, alpha-galactosidase and beta-galactosidase used in this study showed different inhibition profiles against various pericosine derivatives. These results suggest that the steric configurations of amino acids at the ligand-binding site among these enzymes are significantly different. The relationship among the amino acid sequences of the ligand-binding sites and the inhibitory activities will be rigorously evaluated in a subsequent study.”
Comment#2: The study does not need "some" docking, I would like to see a fully detailed analysis of the predicted binding modes. The figures as they appear now do not provide enough insight into how it actually all works. Authors should reduce the zone of visible residues in the active site to the minimum needed for correct understanding of the structure-activity relationships, 2D figures are messy and do not help too much either. I would like to see the correlation of the docked pose scores with the experimental inhibition data measured by the authors or similar compounds measured by other groups.
Response#2: According to reviewer's comment, we corrected on line 19 in page 17 and line 3 in page 7 as “The calculation of the binding affinity scoring function was performed with the amber 10:eht force field, triangle matcher as placement, and GBVI/WSA dG as the binding affinity scoring function[26],” and “Comparisons of affinity scores between different substrates are known to be useful in assessing agonist/antagonist activity [21]. The binding affinity scores of compounds (-)-5, (-)-24, and glucose were -6.8, -4.7, and -8.2 kcal/mol, respectively. Glucose, which is included as a partial residue of natural substrates, exhibited the highest affinity among the three docked compounds. Also, the affinities of compounds (-)-5 and (-)-24 reflected the inhibitory activities, 1.2x10-5 and 1.7x10-3, respectively. These results indicate that the docking studies were achieved with a certain accuracy. On the other hand, the threshold value of the hydrogen bond strength depicted in Figure 2 was set to -1.0 kcal/mol for glucose and -0.1 kcal/mol for derivatives. Compound (-)-5 is considered to contribute to stronger binding affinity by interacting with more peripheral amino acids than compound (-)-24.
Comment#3: the docking protocol should be described in more detail, especially protein preparation with respect to charged residues in the binding site that have a crucial impact on hydrogen bonding and thus on the docked poses and their scoring. According to my pKa prediction program, the GLU 277 residue seems to be protonated in 3A4A which is quite special and may have interesting consequences for docking and activity studies... What authors saw in their case?
Response#3: According to reviewer's comment, we corrected on line 21 in page 17 as “In the protein preparation with respect to charged residues in the binding site, the Protonate3D option in MOE was used to determine the ionization states and add hydrogens to the system[27]. The function of the Protonate 3D allows to assign ionization states and position hydrogens in a macromolecular structure given its 3D coordinates from the crystal structure. Hydrogen atoms are required for all-atom molecular mechanics, dynamics or electrostatic calculations. The addition of hydrogen atoms to a macromolecule is a non-trivial task; generally, one must determine the rotamers of -SH -OH -CH3 and -NH3 groups in CYS, SER, TYR, THR, MET, and LYS; the ionization states of acids and bases in ARG, ASP, GLU, LYS, HIS; the tautomers of imidazoles (HIS) and carboxylic acids (ASP, GLU); the protonation state of metal-ligand atoms CYS, HIS, ASP, GLU, etc.; the ionization state of metals.”
(In our docking conditions, GLU277 was not protonated.)
Reviewer 4 Report
I thank the authors for their answer. I know that many papers are published without inhibition constants, but it does not mean, that it is correct. This leads to gross errors in the evaluation of the results. As I mentioned in my previous note, could the authors at least write one sentence that the conclusions regarding enzyme inhibitory activity drown from IC50 are approximate? For example as mentioned the authors in the answer: Since designed molecules have structural similarity with glucose, so we think these molecules shows the activity by competitive inhibition...and therefore we used IC50 values for the evaluation of the enzyme inhibitory activity.
Author Response
Comment: I thank the authors for their answer. I know that many papers are published without inhibition constants, but it does not mean, that it is correct. This leads to gross errors in the evaluation of the results. As I mentioned in my previous note, could the authors at least write one sentence that the conclusions regarding enzyme inhibitory activity drown from IC50 are approximate? For example as mentioned the authors in the answer: Since designed molecules have structural similarity with glucose, so we think these molecules shows the activity by competitive inhibition...and therefore we used IC50 values for the evaluation of the enzyme inhibitory activity.
Response: we added a sentence above Table 1.
Round 4
Reviewer 2 Report
Authors incorporated my suggestions into their manuscript and improved readability and reproducibility of the modeling part that was criticised by me. The only point where they did not comply was the recommendation to perform MD simulations. If their lab has no suitable computing capability, the study can be published, if the authors explicitly state that their results need to be critically viewed and that MD smilations need to be perfromed.
Author Response
THank you for comment.
Comments: Authors incorporated my suggestions into their manuscript and improved readability and reproducibility of the modeling part that was criticised by me. The only point where they did not comply was the recommendation to perform MD simulations. If their lab has no suitable computing capability, the study can be published, if the authors explicitly state that their results need to be critically viewed and that MD smilations need to be perfromed.
Response: we added a sentence “However, the accuracy of the docking simulation is limited. It is considered necessary to perform MD simulation to obtain more detailed knowledge” in page 7 line 12.